# Data Selection Matters: Towards Robust Instruction Tuning of Large Multimodal Models

**Xu Yang**[1]    **Chen Liu**[† 1]    **Ying Wei**[† 2]
[1] City University of Hong Kong    [2] Zhejiang University
xyang337-c@my.cityu.edu.hk    chen.liu@cityu.edu.hk    ying.wei@zju.edu.cn

## Abstract

Selecting a compact subset of visual instruction–following data has emerged as an effective way to align large multimodal models with human intentions while avoiding the high cost of full-dataset training. Yet we observe that both full-data training and existing state-of-the-art data selection methods tend to inherit underlying dataset biases such as position bias and spurious correlations, leading to biased model behaviors. To address this issue, we introduce ARDS, a robustness-aware targeted visual instruction-selection framework that explicitly mitigates these weaknesses, sidestepping the need for access to downstream data or time-consuming gradient computation. Specifically, we first identify the worst-case evaluation subgroups through visual and textual task-specific perturbations. The robust training mixture is then constructed by prioritizing samples that are semantically closer to these subgroups in a rich multimodal embedding space. Extensive experiments demonstrate that ARDS substantially boosts both robustness and data efficiency for visual instruction tuning. We also showcase that the robust mixtures produced with a smaller model transfer effectively to larger architectures. Our code and selected datasets that have been demonstrated transferable across models are available at https://github.com/xyang583/ARDS.

## 1   Introduction

Large multimodal models (LMMs) [23, 9, 30, 123, 64, 115, 75] have garnered significant attention due to their strong zero-shot capabilities, enabling a wide range of applications in real-world scenarios. Visual instruction tuning [64] has proven effective in enhancing these models' abilities to follow user instructions and reason deeply based on visual cues. Recent efforts have demonstrated that improved performance after visual instruction tuning can be achieved by adjusting image resolutions [59], optimizing training pipelines [75], and scaling up the data used during pre-training and instruction tuning [9, 23, 32, 53, 117, 63, 115].

Despite these advances, existing LMMs often experience significant performance degradation when exposed to minor input perturbations [118, 125]. For instance, as illustrated in Figure 1, LLaVA-1.5 [63], trained on the original dataset, suffers nearly 32% drop in accuracy on the ScienceQA benchmark [70] when answer choices are shuffled or option letters are replaced. Such vulnerabilities are often attributed to dataset biases that inadvertently encourage shortcut learning or spurious correlations [31, 90, 92, 96, 25, 80]. To mitigate the biases, past works have employed post-training strategies such as prompt-based in-context learning [81, 28, 24] or inference-time calibration [118]. However, such methods are limited by researchers' prior and incur additional computational costs during inference. Therefore, we hope to be able to prevent LMMs from learning biased behaviors during visual instruction tuning.

---

[†]Corresponding authors.

39th Conference on Neural Information Processing Systems (NeurIPS 2025).

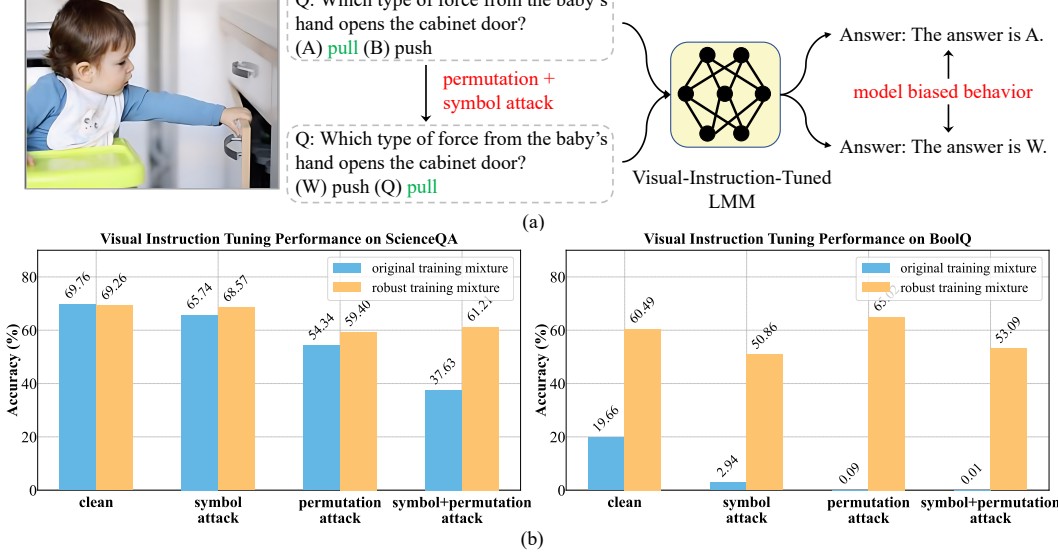

Figure 1: (a) Illustration of biased behaviors exhibited by LMMs. (b) Robustness of visual-instruction-tuned LLaVA-1.5-7B on a multimodal task (left) and LLaVA-1.5-13B on a pure-text task (right) under symbol and permutation attacks. The results highlight a significant decline in accuracy due to underlying dataset biases, and catastrophic forgetting further amplifies the vulnerability. Our curated robust training mixture enhances the model's robustness.

In this paper, we propose constructing a robust training mixture to improve the robustness of visual instruction tuning without modifying the training pipeline or increasing inference costs. The key challenge lies in how to recognize the dataset biases and select training samples that mitigate the vulnerabilities of LMMs. To overcome this challenge, we aim to identify specific training examples that reduce the model's worst-case error. Formally, we introduce a simple but effective targeted data selection method, dubbed as ARDS, to prioritize training on data that contribute most to the worst-case subgroups. Inspired yet different from recent targeted instruction-following data selection methods [107], our method utilizes training sample representations rather than gradients. Technically, we first extract the conversation vector from each training sample to build a vector database. After that, we perform hierarchical clustering to group a holdout set in the embedding space, from which we construct worst-case evaluation subgroups by applying perturbations to identify samples most sensitive to biased model predictions. We then select training samples that are most similar to these specific subgroups. Our method introduces the aforementioned innovations that distinguish it from prior targeted data selection methods [107]: (1) ARDS fully leverages the rich hidden representation of LMMs, avoiding computation-intensive gradient-based information measures, which renders it more efficient than prior methods. (2) Instead of requiring few-shot examples in downstream tasks, ARDS adopts deep clustering and perturbation strategies to maintain the merits of zero-shot generalization capabilities. The overall procedure is summarized in Algorithm 1.

Our contribution can be summarized as follows:

- We propose a gradient-free robustness-aware data selection framework (ARDS) for enhancing the robustness and data efficiency of visual instruction tuning.
- To represent each training sample consisting of multi-turn conversations, the conversation vector is introduced via an attention-score weighted mechanism. The worst-case evaluation subgroups are constructed by clustering and task-specific perturbations, which are leveraged to identify high-quality training samples.
- The robust training mixture curated using a small model demonstrates strong cross-model transferability, significantly improving robustness when applied to training a large model.
- Extensive experiments across eleven evaluation benchmarks validate the effectiveness of our curated robust training mixture. Notably, with only 30% of the training data, ARDS improves robust accuracy by up to 20.62% on multimodal tasks compared to state-of-the-art data selection methods, demonstrating that ARDS can improve both efficiency and robustness.

## 2 Related work

**Visual Instruction Tuning for Large Multimodal Models.** Large multimodal models (LMMs) have witnessed significant advancements and evolved into versatile general-purpose assistants [87, 102, 6, 8, 30, 23, 56, 9, 64, 63, 75, 115]. A key component of the progress is visual instruction tuning, which aligns LMMs to interpret visual content and diverse user prompts while enabling remarkable zero-shot generalization across vision-language tasks [123, 64, 63]. Among existing approaches, LLaVA [64, 63, 54] is distinguished by its data efficiency: whereas Q-former-based models require 129M [23] to 1.4B [9] image–text pairs, LLaVA achieves superior performance with only 665K conversation-style examples [64, 63, 54]. The training examples are carefully transformed and synthesized from established image–text corpora [61, 73, 93, 78, 97] via text-only GPT-4 [2]. The result underscores a key insight of visual instruction tuning: a high-quality instruction dataset can endow an LMM with superior visual perception and reasoning capability, lessening the reliance on ever-larger training corpora [33].

**Data Selection for (Visual) Instruction Tuning.** Instruction-following data selection methods aim to identify a smaller, yet representative, subset of training data that can achieve comparable or even superior performance compared to using the full dataset, demonstrating promising results in improving data efficiency for the alignment of instruction tuning [121, 57, 62, 13, 66, 107, 39, 111, 14, 65, 50, 17, 104, 3, 103] and visual instruction tuning [105, 15, 68, 51, 106]. As shown in Table 1, existing state-of-the-art data selection methods can be categorized based on the information proxy, objective and dependence on targeted downstream task [86, 7]. Based on gradient-based influence function [49, 85, 82], LESS [107] introduces a targeted instruction selection framework, modifying the gradient inner product with cosine similarity to select high-quality subsets responsible for downstream error reduction. Subsequent work eliminates the downstream data requirement with self-influence score [68] and extends to the multi-task setting with majority vote of multiple single task selection [106]. While yielding strong performance, these methods still require computationally expensive gradient information over the full training set. In contrast, we customized targeted instruction selection [107] for improving the robustness against specific dataset biases while avoiding gradient-based drawbacks. Another line of work leverages feature embedding to capture data representation and perform task-agnostic selection based on predefined scores about the model prediction [87, 83, 74], distance to cluster centroids [100], as well as cluster transferability and density [1, 51]. Although effective, they demand a good feature representation space. For example, COINCIDE [51] utilizes a reference model well-trained on the full dataset. In addition, the robustness is overlooked by previous methods, which is the focus of this study. Moreover, our perturbation-based strategy also contributes to the clean performance. We leave the detailed comparisons in Appendix C.

Table 1: Comparisons of existing visual instruction-following data selection methods with large multimodal models. *Information Proxy* indicates the representation used to compute the information measure. *Objective* means the selection goal emphasized when ranking samples. *Task-Aware Selection* denotes methods explicitly target a specific task. *Downstream-Data-free* marks no downstream-task samples are required during selection.

| Method | Information Proxy | Objective | Task-Aware Selection | Downstream-Data-free |
|---|---|---|---|---|
| LESS [107] | Gradient | Quality | ✓ | ✗ |
| ICONS [106] | Gradient | Quality | ✓ | ✗ |
| TIVE [68] | Gradient | Diversity | ✓ | ✓ |
| COINCIDE [51] | Feature | Diversity | ✗ | ✓ |
| **ARDS (Ours)** | Feature | Robustness | ✓ | ✓ |

**Data Selection for Robustness.** Recent studies have shown that deep neural networks often leverage dataset biases for vulnerable high performance on specific tasks, such as visual spurious features [88, 31, 99], language spurious association [36], position and stereotype bias [92, 80]. There is a series of works that leverage data selection to enhance the model's robustness and mitigate biased behavior before training or after training. For example, DoReMi [108] trains a proxy model using Group DRO [89] and prioritizes training on valuable and learnable samples with higher excess loss of the proxy model (i.e., the difference between training and validation loss). [24] proposes a causal-guided method to recognize samples that contain biases, which are used as negative examples to suppress bias behavior via in-context learning. Most related to our work is D3M [44], which leverages TRAK [82] to remove specific examples by measuring gradient-based influence on the worst-group loss. However, our gradient-free approach enables extremely efficient and robust data selection for large multimodal

models. To the best of our knowledge, we are the first to leverage data selection for improving the large multimodal model's robustness against spurious correlation and position bias.

## 3   Preliminary

**Problem Formulation.**   We focus on improving both data efficiency and robustness in visual instruction tuning for large multimodal models. After tuning, the model's zero-shot generalization capability is evaluated on clean as well as perturbed inputs across downstream tasks. This requires the model to consistently interpret the visual cues and follow user instructions to answer the question. During training, the model is optimized on a training corpus $\mathcal{D}$ consisting of multi-turn conversations $\boldsymbol{x} = \{\boldsymbol{x}_i\}_{i=1}^T$, where $T$ is the total number of turns and $\boldsymbol{x}_i = \left( \boldsymbol{x}_i^{\text{img}}, \boldsymbol{x}_i^{\text{ins}}, \boldsymbol{x}_i^{\text{ans}} \right)$, where $\boldsymbol{x}_i^{\text{img}}$, $\boldsymbol{x}_i^{\text{ins}}$, and $\boldsymbol{x}_i^{\text{ans}}$ denote the image, instruction, and answer in each conversation $\boldsymbol{x}_i$, respectively. We hypothesize that vulnerabilities of model behaviors, such as *position bias* and *spurious correlation*, arise from underlying dataset biases. Thus, we address this issue via a simple but effective data selection strategy. Formally, given a target model $f_\phi$ with parameters $\phi$, our goal is to build a robust training mixture guided by a proxy model $f_\theta$ with parameters $\theta$, where $\|\theta\| = \|\phi\|$ or even $\|\theta\| \ll \|\phi\|$. Our selection objective is to minimize the worst-case error of $f_\phi$ on downstream tasks.

**Visual Instruction Tuning.** LLaVA [64, 63] proposes two-stage instruction-following alignment procedure. In the vision-language alignment pretraining stage, an MLP cross-modal connector is trained to project the visual features extracted by the CLIP encoder to the language embedding space. After that, in the visual instruction tuning stage, the decoder layers of the large language model are optimized in a supervised fine-tuning manner. The training objective is auto-regressive teacher forcing with the cross-entropy loss $\ell$ forecasting the $Q$ answer tokens.

$$\ell = -\frac{1}{Q} \sum_{i=1}^{Q} \log P_\theta \left( \boldsymbol{x}_i \mid \boldsymbol{x}_i^{\text{img}}, \boldsymbol{x}_i^{\text{ins}}, \boldsymbol{x}_i^{\text{ans}}, _{<i} \right). \tag{1}$$

**Targeted Data Selection.** Inspired by the influence function [37, 49], LESS [107] builds the gradient datastore by calculating gradients of all trainable parameters for each training sample $g_{tr}^i = \frac{\partial \ell_i}{\partial \theta}$, where $\ell_i$ denotes the average loss of all tokens of $i$-th training sample using the proxy model $f_\theta$. The influence of a training sample $x_i$ is quantified by the cosine similarity between its gradient and the gradients of the downstream few-shot examples and the maximum similarity over all few-shot examples as its final influence score

$$\mathcal{I}(x_i) = \max_j \cos \left( g_{tr}^i; g_{te}^j \right), \tag{2}$$

where $g_{te}^j$ denotes the gradient of the $j$-th downstream few-shot example.

## 4   Methods

The schematic illustration of our adversarial representation-based data selection is demonstrated in Figure 2. In Sec. 4.1, we present the whole of our data selection pipeline. Then we detail each step of our proposed method in Sec. 4.2.

### 4.1   Overview

We aim to improve the robustness of large multimodal models after visual instruction tuning while minimizing the required amount of training data. Two key criteria are enforced in the building of a robust training mixture. First, the selection proxy used to determine which samples to retain should effectively preserve the most substantial and representative information from the original data. Selecting irrelevant or uninformative samples could even exacerbate existing dataset biases or spurious correlations in downstream tasks. Second, the data selection algorithm should avoid excessive computational overhead. Pursuing training efficiency with fewer samples via computation-intensive selection strategies (e.g., relying on full-parameter gradients) might make the proposed algorithm impractical, particularly for large multimodal models, which typically encounter each

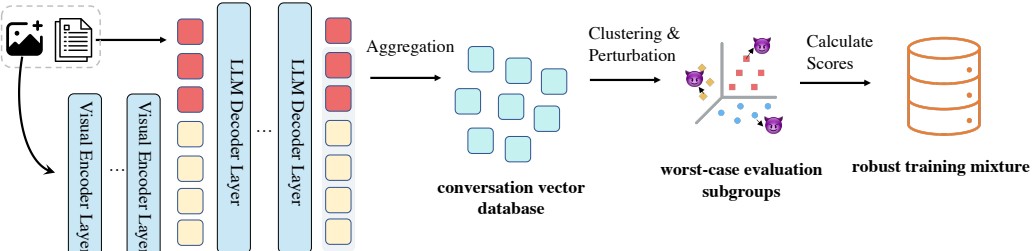

Figure 2: **ARDS Pipeline:** (1) **Conversation Vector Database**: ARDS begins with extracting token embeddings for each conversation and then aggregates the substantial information within the conversation based on attention scores. (2) **Worst-case Evaluation Subgroups**: The hierarchical clustering is performed in the embedding space. The perturbation is applied to find representative samples that are most vulnerable to model-biased behavior. (3) **Quality Score Measure**: The training sample scores are measured by distance with each worst-case subgroup. Subsequently, the highest-scoring samples are selected to curate the robust training mixture.

training sample only once during fine-tuning with one epoch [63]. To resolve these challenges, we propose to leverage hidden representations as lightweight yet expressive proxies for measuring the contribution of each training sample. The key intuition is to construct a robust training mixture that prioritizes training samples semantically similar to vulnerable samples in the worst-case subgroups. To this end, we first build a vector database by extracting the conversation vector for each training sample and then curate worst-case evaluation subgroups to evaluate potential biases. Finally, the large multimodal models are instruction-tuned on the constructed robust training mixture to avoid biased behaviors and enhance robustness against identified vulnerable subpopulations.

## 4.2 Data Selection for Robust Visual Instruction Tuning

**Conversation Vector Database.** Large multimodal models are often trained on samples consisting of multi-turn conversations. To represent each training sample, we introduce an attention-score weighted mechanism that aggregates the conversation vector from the token-level embeddings based on their relevance. Formally, for an input with $L$ tokens, we first extract token embeddings $\mathbf{H}$ and then evaluate the importance of previous visual and textual tokens by examining the attention score. Specifically, in the multi-head self-attention block of a typical transformer, the attention score of each head $h$ is calculated by $\mathbf{A}_h = \mathbf{Q}_h \mathbf{K}_h^\top / \sqrt{d_k}$, where $\mathbf{Q}_h$ and $\mathbf{K}_h$ are the query and key embeddings, respectively. $d_k$ is the head dimension. We then obtain the average attention score across heads $\tilde{\mathbf{A}} \in \mathbb{R}^{L \times L}$. Let $\mathbf{H}_t \in \mathbb{R}^{1 \times d}, t \in \{1, \cdots, L\}$ represent the token hidden states at the final embedding layer of the proxy model, where $L$ is the sequence length and $d$ is the hidden dimension. For each sample, the attention scores corresponding to tokens before the last token are extracted, i.e., $\mathbf{A}_L \in \mathbb{R}^{1 \times (L-1)}$, each element represents the relative importance of a token in the sequence. The visual and textual tokens preceding the last token are then aggregated through the weighted sum, $\hat{\mathbf{H}} = \sum_{t=1}^{L-1} \mathbf{A}_{L,t} \cdot \mathbf{H}_t$. The final conversation vector $r$ for each sample is obtained by concatenating the last token and the aggregated token:

$$r = [\mathbf{H}_L; \hat{\mathbf{H}}], \tag{3}$$

where $r \in \mathbb{R}^{2 \times d}$. The design of the conversation vector leverages both the information from the last token and the contextual relationships between tokens, as encoded by the attention mechanism, providing a robust and rich representation for each training sample. Unlike gradient-based data selection methods involving both forward and backward computation, our method needs only one forward to build the vector database over the training corpus.

**Worst-case Evaluation Subgroups.** We first partition the training set into semantically coherent subgroups via hierarchical clustering over the instruction embeddings. Specifically, using the established conversation vector database, we perform spherical $K$-means [5] clustering to yield $M$ subgroups. Let $\mathcal{C}_m$ denote the $m$-th subgroup, where $m \in \{1, \ldots, M\}$. After clustering, we aim to find samples that are most susceptible to the model's biased behavior. Drawing inspiration from the crucial role of support vectors shaping the decision boundary [21, 40], we aim to apply perturbation to

identify vulnerable samples near the decision boundary. Concretely, we impose diffusion noise [122] on the image, which is more easily to trigger different or erroneous predictions than other visual corruptions [124, 122]. For the textual variations, we inject task-aware perturbations designed to improve robustness against specific attacks. To this end, we apply dual perturbations to obtain the corrupted conversation $x'$. We then retrieve $\text{top}_B$ conversations with the largest loss difference for each cluster, i.e.,

$$\mathcal{S}_m = \text{top}_B \{\mathbf{x} \in \mathcal{C}_m : |\ell(\mathbf{x}) - \ell(\mathbf{x}')|\}, \tag{4}$$

where $\ell$ is the cross entropy loss as Eq.(1) and $\text{top}_B$ denotes the operation of selecting $B$ highest-scoring samples. We can then leverage the balanced worst-case subgroups $\mathcal{S} = \{\mathcal{S}_1, \cdots, \mathcal{S}_M\}$ to quantify the model bias by model performance degradation. To evaluate the difficulty for each worst-case subgroup, we use the average loss $\ell_{\mathcal{S}_m} = \frac{1}{B} \sum_{j \in \mathcal{S}_m} \ell_j$. Note that we only incorporate perturbations to construct worst-case evaluation subgroups and do not apply any intervention on the original training data.

**Robust Coreset Selection.** To quantify the importance of each training sample in reducing biased behaviors, we aim to capture the extent to which training samples contribute most to improving the worst-case performance. Unlike the previous method [44] that removes training samples leading to high worst-case error via gradient information, we assess training sample quality based on its distance with our built worst-case evaluation subgroups in the embedding space. Specifically, for the $i$-th training sample and $m$-th worst-case evaluation subgroup, we first measure the cosine similarity between their conversation vectors,

$$d_{i\mathcal{S}_m} = \frac{1}{B} \sum_{j \in \mathcal{S}_m} \cos\left(r_{\text{tr}}^i; r_{\mathcal{S}_m}^j\right). \tag{5}$$

Then, we follow [44] to use softmax-weighted aggregation to obtain the information value score as below

$$\mathcal{I}(x_i) = \frac{\sum_{m=1}^{M} \exp(\ell_{\mathcal{S}_m}) \cdot d_{i\mathcal{S}_m}}{\sum_{m=1}^{M} \exp(\ell_{\mathcal{S}_m})}. \tag{6}$$

**Robust Training Mixture.** After the quality score measure, we select training conversations with the highest scores to build a robust training mixture $\mathcal{D}_{\text{robust}}$. The target model $f_\phi$ is fine-tuned on the $\mathcal{D}_{\text{robust}}$ using the training objective Eq.(1). The complete algorithm is shown in Algorithm 1 of Appendix I. We provide the theoretical analysis in Appendix J.

## 5 Experiments

In this section, we first elaborate on the experimental setup in Section 5.1. Then we compare our proposed ARDS with the latest state-of-the-art methods to evaluate the effectiveness in Section 5.2. Additionally, we present ablation studies in Section 5.3 and conduct a broader analysis in Section 5.4.

### 5.1 Experimental Setup

**Implementation Details.** We use the original training corpus LLaVA-665K [63] for our robust training mixture curation without introducing any external data. After data selection, visual instruction tuning is performed with the same training configuration for all instruction selection baselines. Following LESS [107], we use a warmed-up LLaVA-1.5 (7B) model as our proxy model for data selection, which is trained on the subset of data with randomly sampled 1000 examples for 4 epochs. We set the number of clusters $K$ to 70 and the subgroup budget $B$ to 50. When building the worst-case evaluation subgroups and measuring robustness during test time, the adversarial permutation attack (PA) is injected by generating all possible permutations via brute-force algorithms (i.e., $k!$ permutations, where k is the number of options) and the symbol attack (SA) is incorporated through replacing the standard symbols used in answer choices (e.g., A/B/C/D) with a different set of characters (e.g., Q/W/E/R). We refer to Appendix D for a more detailed description of selection, training, and evaluation. We conduct a comprehensive hyperparameter study in Appendix E.

**Baselines.** We compare ARDS with several baseline methods. Specifically, the random sampling selection, denoted by **Random**, randomly selects a subset from the original training mixture. We also compare with the gradient-based targeted selection method **LESS** [107], which leverages downstream

Table 2: **Zero-shot robust accuracies (%, ↑)** against spurious correlation and position bias. ARDS and the baselines select the same size of training data for the visual instruction tuning of LLaVA-1.5 (7B) [63] with the same training configurations. **SA**: symbol attack. **PA**: permutation attack. For each evaluation task, results surpassing **Full** method are highlighted in **bold** and the optimal result achieved among all curated training mixtures is underlined.

| Selection Method | Data Percentage | ScienceQA Clean | PA | SA | SA + PA | Avg. | SEED-Bench Clean | PA | SA | SA + PA | Avg. | MMBench-EN Clean | PA | SA | SA + PA | Avg. | MMBench-CN Clean | PA | SA | SA + PA | Avg. |
|---|---|---|---|---|---|---|---|---|---|---|---|---|---|---|---|---|---|---|---|---|---|
| Full | 100% | 69.76 | 54.34 | 65.74 | 37.63 | 56.87 | 59.65 | 41.92 | 54.83 | 22.40 | 44.69 | 74.84 | 61.15 | 69.39 | 41.09 | 61.62 | 69.95 | 52.34 | 65.33 | 34.90 | 55.63 |
| Random | 30% | 69.76 | 52.60 | 59.44 | 23.75 | 51.39 | 56.84 | 35.74 | 46.58 | 12.73 | 37.97 | 74.20 | 57.75 | 65.49 | 31.83 | 57.32 | 69.76 | 49.50 | 63.78 | 34.33 | 54.34 |
| LESS-SciQA [107] | 30% | 68.42 | 55.63 | 64.70 | 34.95 | 55.93 | 55.82 | 36.30 | 52.32 | 18.19 | 40.66 | 72.14 | 57.89 | 67.54 | 34.51 | 58.02 | 67.38 | 48.49 | 62.05 | 30.68 | 52.15 |
| RHO-LOSS [76] | 30% | 64.01 | 36.89 | 59.44 | 21.42 | 45.44 | 53.97 | 25.07 | 48.36 | 11.26 | 34.67 | 70.82 | 49.90 | 66.94 | 32.83 | 55.12 | 68.05 | 43.68 | 65.03 | 31.90 | 52.16 |
| COINCIDE [51] | 30% | 67.72 | 52.21 | 61.08 | 28.06 | 52.27 | 57.49 | 36.02 | 48.93 | 15.88 | 39.58 | 73.78 | 58.65 | 68.10 | 37.65 | 59.54 | 69.48 | 49.64 | 64.84 | 35.97 | 54.98 |
| ARDS (ours) | 30% | 69.26 | **59.40** | **68.57** | **47.60** | **61.21** | 58.11 | 40.73 | 56.83 | 31.52 | 46.80 | 74.43 | 61.03 | 72.37 | 53.22 | 65.26 | 70.48 | 53.73 | 68.98 | 46.02 | 59.80 |

| Selection Method | Data Percentage | A-OKVQA Clean | PA | SA | SA + PA | Avg. | MMMU Clean | PA | SA | SA + PA | Avg. | ARC-e Clean | PA | SA | SA + PA | Avg. | BoolQ Clean | PA | SA | SA + PA | Avg. |
|---|---|---|---|---|---|---|---|---|---|---|---|---|---|---|---|---|---|---|---|---|---|
| Full | 100% | 80.52 | 72.31 | 78.34 | 55.02 | 71.54 | 35.06 | 10.15 | 33.65 | 4.84 | 20.92 | 36.76 | 11.11 | 25.25 | 0.83 | 18.48 | 37.77 | 23.64 | 4.53 | 0.09 | 16.50 |
| Random | 30% | 78.25 | 66.29 | 70.13 | 35.72 | 62.59 | 34.00 | 9.21 | 35.77 | 5.43 | 21.10 | 38.95 | 12.38 | 33.99 | 1.36 | 21.67 | 55.93 | 29.79 | 37.22 | 3.39 | 31.58 |
| LESS-SciQA [107] | 30% | 78.60 | 66.72 | 74.41 | 45.94 | 66.42 | 37.43 | 11.81 | 33.53 | 4.49 | 21.82 | 37.86 | 13.57 | 35.18 | 3.03 | 22.41 | 57.58 | 40.86 | 39.36 | 3.27 | 35.27 |
| RHO-LOSS [76] | 30% | 76.86 | 55.02 | 71.00 | 37.64 | 60.13 | 34.00 | 5.31 | 32.23 | 3.19 | 18.68 | 38.21 | 5.49 | 34.39 | 1.27 | 19.84 | 43.79 | 41.37 | 37.80 | 0.61 | 22.65 |
| COINCIDE [51] | 30% | 77.55 | 65.59 | 72.66 | 44.10 | 64.97 | 37.90 | 9.80 | 33.29 | 3.54 | 21.13 | 38.25 | 11.86 | 36.06 | 2.64 | 22.20 | 55.14 | 29.20 | 41.01 | 5.20 | 32.63 |
| ARDS (ours) | 30% | 78.34 | 71.09 | 77.64 | 64.72 | 72.95 | 37.54 | 12.75 | 34.24 | 6.97 | 22.88 | 39.92 | 16.95 | 37.15 | 8.26 | 25.57 | 58.62 | 46.45 | 46.85 | 17.25 | 42.29 |

labeled few-shot examples for task-specific selection. Note that we use the same proxy model to build the gradient store in LESS and our conversation vector database. Moreover, we compare our method with the **RHO-LOSS** [76] implemented by directly training the proxy model on our holdout worst-case subgroups to score training data with the excess loss [76, 108, 62]. For the recent state-of-the-art coreset selection method **COINCIDE** [51], we reproduce its best version leveraging the TinyLLaVA-2B [120] as the proxy model, which has already been well-trained on full training data. The implementation details of the baseline methods are elaborated in Appendix D.

## 5.2 Comparison with Baselines

**Effectiveness of the robust training mixture.** We evaluate visual instruction tuning on the mixture with the size of 30 % training data selected by ARDS and by several baselines. As Table 2 shows, ARDS consistently outperforms all competitors on a diverse set of downstream evaluation tasks Concretely, ARDS yields the largest robustness gains while matching the full-data model on the clean performance for multimodal input (e.g., MMMU, A-OKVQA), pure text input (e.g., ARC, BoolQ) and cross-lingual tasks (e.g., MMBench-CN), underscoring its generality. Compared to representative task-aware selector LESS [107], our method delivers higher robustness and data-efficiency without requiring downstream few-shot examples or expensive gradient information. Moreover, recent state-of-the-art task-agnostic method COINCIDE [51] remains susceptible to symbol and permutation attacks. The performance of baseline RHO-LOSS [76] is less than satisfactory, which uses a proxy model trained on the worst-case subgroups to score examples via the excess loss, probably because fitting an optimal large multimodal model on the small held-out set is intrinsically hard. In contrast, ARDS removes the biased samples and constructs the robust training mixture by utilizing worst-case evaluation subgroups to pay more attention to samples near the decision boundary.

Table 3: **Cross-Architecture-Scale Transferability.** Our robust training mixture curated via LLaVA-1.5 (7B) generalizes effectively to larger architectures, inducing strong robustness improvement when trained on selected data for LLaVA-1.5(13B).

| Proxy Model | Target Model | Selection Method | Data Percentage | ScienceQA Clean | PA | SA | SA + PA | Avg. | SEED-Bench Clean | PA | SA | SA + PA | Avg. | MMBench-EN Clean | PA | SA | SA + PA | Avg. | MMBench-CN Clean | PA | SA | SA + PA | Avg. |
|---|---|---|---|---|---|---|---|---|---|---|---|---|---|---|---|---|---|---|---|---|---|---|---|
| - | LLaVA-1.5 (13B) | Full | 100% | 71.05 | 57.21 | 64.20 | 37.58 | 57.51 | 61.12 | 43.85 | 56.19 | 23.08 | 46.06 | 76.02 | 64.06 | 71.73 | 47.79 | 64.90 | 72.88 | 57.36 | 68.68 | 37.77 | 59.17 |
| - | LLaVA-1.5 (13B) | Random | 30% | 70.25 | 54.69 | 63.76 | 31.33 | 55.01 | 59.08 | 39.06 | 52.09 | 16.17 | 41.60 | 75.70 | 59.92 | 69.74 | 39.50 | 61.22 | 72.23 | 53.98 | 65.28 | 31.74 | 55.81 |
| LLaVA-1.5 (7B) | LLaVA-1.5 (13B) | ARDS (ours) | 30% | 72.58 | 60.19 | 66.14 | 41.99 | 60.22 | 59.94 | 43.98 | 57.58 | 30.76 | 48.07 | 76.41 | 64.24 | 72.95 | 52.60 | 66.55 | 71.49 | 56.18 | 67.45 | 40.06 | 58.80 |

| Proxy Model | Target Model | Selection Method | Data Percentage | A-OKVQA Clean | PA | SA | SA + PA | Avg. | MMMU Clean | PA | SA | SA + PA | Avg. | ARC-e Clean | PA | SA | SA + PA | Avg. | BoolQ Clean | PA | SA | SA + PA | Avg. |
|---|---|---|---|---|---|---|---|---|---|---|---|---|---|---|---|---|---|---|---|---|---|---|---|
| - | LLaVA-1.5 (13B) | Full | 100% | 82.36 | 73.28 | 80.70 | 62.88 | 74.80 | 38.25 | 14.29 | 35.77 | 6.14 | 23.61 | 18.36 | 0.53 | 14.58 | 0.09 | 8.39 | 19.66 | 2.94 | 0.09 | 0.01 | 5.68 |
| - | LLaVA-1.5 (13B) | Random | 30% | 79.74 | 69.61 | 77.21 | 50.22 | 69.19 | 38.84 | 12.63 | 35.42 | 4.72 | 22.90 | 45.63 | 17.35 | 41.37 | 7.51 | 27.96 | 56.57 | 39.89 | 63.09 | 40.67 | 50.06 |
| LLaVA-1.5 (7B) | LLaVA-1.5 (13B) | ARDS (ours) | 30% | 80.96 | 72.66 | 79.83 | 63.41 | 74.22 | 40.50 | 15.94 | 38.37 | 8.74 | 25.89 | 45.98 | 22.57 | 42.07 | 12.12 | 30.69 | 60.49 | 50.86 | 65.02 | 53.09 | 57.37 |

**Scaling the model.** To test cross-model-scale transfer, we fine-tune the larger LLaVA-1.5 (13B) on the robust training mixture curated with the approximately 2x smaller proxy model LLaVA-1.5(7B). The results in Table 3 demonstrate that ARDS effectively transfers the curated robust training mixture from weaker models to larger, more powerful models. For example, with only 30% of the training data, ARDS enhances the target model's robustness against the strongest symbol plus permutation attacks by 1.93% and 3.70% on ScienceQA and SEED-Bench, respectively. We also observe that train-test modality mismatch further exacerbates the model's vulnerability under input variations, especially for larger models on text-only benchmarks, which may result from catastrophic forgetting [116, 60].

The performance of Vicuna-v1.5-13B [119] on BoolQ gradually degrades as the amount of visual instruction tuning data increases (see Appendix H.8). Interestingly, simple random selection alleviates this degradation, and the mixture produced by ARDS surpasses it and delivers the largest robustness gains. The results highlight that our robustness-oriented selection is essential for reliably scaling LMMs.

**Scaling the data.** We next examine how ARDS behaves across varying training mixture sizes in Figure 3. Randomly discarding samples provides no robustness benefit and often incurs a loss. In contrast, the robust mixtures selected by ARDS deliver consistently higher robustness at every budget. See Appendix F for more figures.

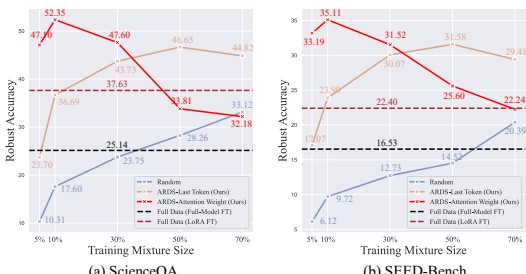

Figure 3: Robust Accuracies (↑) across different sizes of training mixture used for visual instruction tuning of LLaVA-1.5 on ScienceQA and SEED-Bench benchmarks. Robustness is evaluated through incorporating symbol and adversarial permutation attacks.

**Improving robustness against visual spurious correlation.** We further evaluate whether ARDS mitigates biases that arise when particular visual attributes or textual descriptions dominate the training corpus, creating spurious shortcuts [88, 31, 90, 58]. A classic example is a model that tends to answer "yellow" for any banana, even when the image shows a green one.

Following the protocol of [47], we construct the robust training mixture targeted for GQA-OOD, which defines OOD samples as infrequent events and creates fine-grained shifts by extracting questions from the most imbalanced answer groups of GQA [42]. We build our worst-case subgroups using the validation split (See Appendix D for more details). As Table 4 shows, visual instruction tuning with our ARDS achieves

Table 4: **Zero-shot robust accuracies (%, ↑) against visual spurious correlation**.

| Selection Method | Data Percentage | GQA | | | | |
|---|---|---|---|---|---|---|
| | | Original | OOD-All | OOD-Head | OOD-Tail | Avg. |
| **Full** | 100% | 61.94 | 57.51 | 61.17 | 51.55 | 58.04 |
| **Random** | 50% | 60.69 | 55.97 | 60.30 | 48.92 | 56.47 |
| **COINCIDE** | 50% | 61.88 | 56.58 | 60.30 | 50.52 | 57.32 |
| **ARDS (ours)** | 50% | 62.43 | 58.44 | 62.26 | 52.21 | 58.84 |

the highest accuracy and robustness against the subpopulation shifts on OOD tasks, surpassing both the random-sampling baseline and COINCIDE [51], a state-of-the-art diversity-oriented selector.

## 5.3 Ablation Study

**Effectiveness of conversation vector.** We compare our attention-score-weighted conversation vector with alternative vectorization strategies. A prevalent baseline in retrieval-augmented generation (RAG) is the last-token embedding, which captures aggregated semantic information [113, 112]. However, the single last token often overlooks rich contextual cues from preceding turns, particularly in multi-turn or multimodal settings. In contrast, our approach leverages the attention map to re-weight all preceding visual and textual token embeddings according to their learned importance. The design explicitly captures which parts of the conversation the model attends to when forming its final prediction, thereby preserving cross-turn dependencies and text–vision interactions that the last-token embedding fails to encode. To ensure a fair comparison, we re-implemented this baseline to build the vector database and reran our selection pipeline. As shown in Table 5, the attention-weighted conversation vector better preserves more significant and useful semantics and achieves higher worst-case performance across benchmarks, demonstrating a clear advantage over the simpler last-token representation, especially under limited training data budget.

Table 5: Ablation study results comparing conversation vector variants.

| Conversation Vector | Data Percentage | ScienceQA | | | | | SEED-Bench | | | | | MMBench-EN | | | | | A-OKVQA | | | | |
|---|---|---|---|---|---|---|---|---|---|---|---|---|---|---|---|---|---|---|---|---|---|
| | | Clean | PA | SA | SA + PA | Avg. | Clean | PA | SA | SA + PA | Avg. | Clean | PA | SA | SA + PA | Avg. | Clean | PA | SA | SA + PA | Avg. |
| Last Token | 10% | 66.68 | 54.04 | 64.06 | 36.69 | 55.36 | 55.13 | 35.02 | 52.29 | 23.90 | 41.58 | 73.11 | 57.61 | 71.06 | 49.53 | 62.82 | 76.33 | 64.45 | 74.59 | 54.32 | 67.42 |
| Attention Weight | 10% | 69.66 | 55.88 | 69.21 | 52.35 | 61.78 | 53.86 | 38.30 | 53.38 | 35.11 | 45.16 | 72.07 | 59.90 | 71.56 | 57.13 | 65.16 | 77.90 | 70.04 | 76.94 | 66.72 | 72.90 |
| Last Token | 30% | 69.41 | 57.36 | 65.99 | 43.73 | 59.12 | 58.23 | 41.47 | 56.12 | 30.07 | 46.47 | 75.84 | 62.09 | 73.48 | 53.22 | 66.15 | 78.95 | 71.18 | 78.08 | 63.32 | 72.88 |
| Attention Weight | 30% | 69.26 | 59.40 | 68.57 | 47.60 | 61.21 | 58.11 | 40.73 | 56.83 | 31.52 | 46.80 | 74.43 | 61.03 | 72.37 | 53.22 | 65.26 | 78.34 | 71.09 | 77.64 | 64.72 | 72.95 |

**The effect of worst-case evaluation subgroups.** ARDS builds worst-case evaluation subgroups through deep clustering to group semantically similar conversations, followed by dual perturbations to identify samples near the decision boundary. To demonstrate the necessity of each component, we conduct the ablation study by removing the components one by one in Table 6. Specifically, in the first

row, we randomly sample the same number of samples $MB$ as the final total size of the worst-case subgroups, and aggregate the importance score of each training data via a sample-to-sample weighted sum, formally given by $\mathcal{I}(x_i) = \frac{\sum_{j=1}^{MB} \exp(\ell_j) \cdot d_{ij}}{\sum_{j=1}^{MB} \exp(\ell_j)}$. For the second row, we apply Eq.(4) to directly retrieve top-$MB$ samples from the training dataset with the largest loss difference, and use the sample-to-sample weighted-sum rule to obtain the importance score. The third row denotes our final construction strategy, which first performs clustering and then retrieves top-$B$ candidates from each cluster via Eq.(4). The importance scores are aggregated using the sample-to-cluster weighted-sum scheme defined in Eq.(6). Eliminating either component markedly reduces robustness, demonstrating the benefits of clustering and perturbation strategies in discovering diverse and challenging subgroups. Furthermore, a finer-grained ablation in Table 11 of Appendix G highlights the effectiveness of visual and textual perturbations for the robustness gains.

Table 6: Ablation study results comparing different components for worst-case evaluation subgroups. LLaVA-1.5 (7B) is utilized as the proxy and target model for all methods.

| Worst-case Evaluation Subgroup | | Data | ScienceQA | | | | | SEED-Bench | | | | | MMBench-EN | | | | | A-OKVQA | | | | |
| Perturbation | Clustering | Percentage | Clean | PA | SA | SA + PA | Avg. | Clean | PA | SA | SA + PA | Avg. | Clean | PA | SA | SA + PA | Avg. | Clean | PA | SA | SA + PA | Avg. |
|---|---|---|---|---|---|---|---|---|---|---|---|---|---|---|---|---|---|---|---|---|---|---|
| ✗ | ✗ | 30% | 65.29 | 51.66 | 62.17 | 30.44 | 52.39 | 56.75 | 35.40 | 51.01 | 18.56 | 40.43 | 73.76 | 57.80 | 69.14 | 41.44 | 60.53 | 77.12 | 65.33 | 74.06 | 47.60 | 66.03 |
| ✓ | ✗ | 30% | 67.43 | 54.34 | 64.35 | 36.49 | 55.65 | **58.38** | 40.42 | 56.24 | 26.57 | 45.40 | 74.15 | 60.89 | 71.96 | 49.36 | 64.09 | **79.04** | 70.92 | 76.77 | 59.56 | 71.57 |
| ✓ | ✓ | 30% | **69.26** | **59.40** | **68.57** | **47.60** | **61.21** | 58.11 | **40.73** | **56.83** | **31.52** | **46.80** | **74.43** | **61.03** | **72.37** | **53.22** | **65.26** | 78.34 | **71.09** | **77.64** | **64.72** | **72.95** |

**Score aggregation strategy.** We conduct an ablation study comparing two score aggregation approaches in our robustness-aware selection. The Subgroup Maximum strategy follows the Equation 2 used in targeted instruction tuning [107], where the information score $\mathcal{I}(x_i)$ is computed as the maximum cosine similarity between a training sample and any single worst-case subgroup. The Subgroup Weighted Sum approach considers both subgroup similarity and subgroup difficulty. See Appendix G for more details. As shown in Table 10 of Appendix G, the weighted sum strategy outperforms the maximum-based aggregation, yielding higher robust accuracy under more challenging attack settings. This suggests that incorporating subgroup difficulty helps select training samples that more effectively target model-biased behaviors.

## 5.4 More Analysis

**Data Selection Computational Analysis.** To conduct the computational analysis, we first compare with task-aware selection baselines LESS [107], using the same LLaVA-1.5 (7B) warmed up for four epochs for a fair comparison. As shown in Table 7, LESS extracts LoRA gradients for each training sample along the discrete training trajectory, which is computationally expensive. By contrast, ARDS encodes each conversation with only the final-layer embedding, completely avoiding per-sample gradient calcu-

Table 7: **Data selection clock time comparison**. All experiments are conducted on 8 Nvidia RTX A6000 GPUs. **Stage1**: Warmup training of the proxy model. **Stage2**: Gradient store or conversation database is built. **Stage3**: Sample quality measure.

| Method | Stage1 | Stage2 | Stage3 |
|---|---|---|---|
| LESS [107] | 8min | 128h | 1min |
| ARDS (Ours) | 8min | 30min | 1min |

lations and cutting the wall-clock time by a wide margin. Furthermore, our runtime is comparable to COINCIDE, which extracts representations from five layers in roughly 100 minutes.

**The impact of LoRA parameter-efficient fine-tuning.** We use Low-Rank Adaptation (LoRA) [41] for parameter-efficient visual instruction tuning in our main experiments. We here study the impact of this training approach on the robustness of large multimodal models. As shown in Table 12 of Appendix H.1, we observe that LoRA tuning only 4.6% of the parameters achieves better robustness with comparable clean performance. This outcome aligns with the findings in [16].

**Generalization to More Challenging Benchmarks.** We further evaluate the generalization capability of our curated robust training mixture on several more challenging benchmarks, especially two math-related visual reasoning benchmarks, MathVista [69] and DynaMath [126]. These two mathematical benchmarks represent more difficult Out-of-Domain (OOD) tasks, as the original LLaVA–665K dataset does not contain explicit mathematical training data [63] (See Appendix H.2 for more details.) As shown in Table 13 of Appendix H.2, our curated robust training mixture consistently enhances average performance across OOD tasks for visual instruction tuning, outperforming both full-data training and previous state-of-the-art data selection strategies.

**Generalization to Other Large Multimodal Models.** To assess the generality and effectiveness of ARDS, we further conduct a transferability study across a range of backbone architectures beyond the Vicuna-based LLaVA-1.5. Specifically, we apply ARDS to two representative state-of-the-art large multimodal models, LLaVA-1.6-Mistral [45] (denoted LLaVA-Mistral) and Qwen2.5-VL-Instruct [10] (denoted Qwen2.5-VL). We refer to Appendix H.4 for more training details. As summarized in Table 15 of Appendix H.4, ARDS consistently enhances robustness and data efficiency across architectures, yielding clear gains in both visual instruction tuning and post-training settings. The results collectively demonstrate the adaptability of ARDS and can be easily applied to various large multimodal models to achieve consistent robustness improvements.

**Generalization under Unseen Attacks and Visual Corruptions.** We here take a further step to investigate the robustness and generalization of our robust training mixture. We introduce two previously unseen symbol attacks and an additional visual corruption. Specifically, the canonical answer labels A/B/C/D are replaced by S/N/V/F and U/I/O/P, respectively, thereby injecting novel variations not encountered during data selection. Note that the used adversarial permutation attack is adaptively generated for each new question by enumerating all possible permutations of the answer options. As shown in Table 17 of Appendix H.6, ARDS consistently achieves strong robustness and generalization even in the presence of these new, previously unseen attacks. Even when compared with the COINCIDE+worst-case subgroup, ARDS maintains superior robust accuracy across all scenarios, confirming that the robustness gains generalize well beyond attacks seen during data selection. Moreover, the stronger diffusion noise is injected into the test images to simulate corrupted visual input. The results in Table 16 of Appendix H.6 show that our ARDS maintains the highest robust accuracy across all scenarios, confirming that our robustness gains are not fully confined to the attacks used during selection.

**Complementary with Data Augmentation.** We investigated whether lightweight data-augmentation schemes can eliminate dataset biases. To mitigate symbol-content spurious correlations and position biases, we apply random symbol replacement and option shuffling during visual instruction tuning. To address visual spurious correlation and subpopulation shifts, we apply AutoAugment [22] to the training images. As shown in Table 18 of Appendix H.7, though textual-level augmentations improve robustness against seen perturbations, they fail to generalize to unseen attacks. Furthermore, simple image-level augmentation does not bring noticeably higher accuracy on shifted subgroups and even degrades clean accuracy on GQA. The observations are consistent with recent findings that naive augmentations often preserve the statistical properties of the original data and can inadvertently amplify existing biases rather than mitigate them [11, 95, 79]. In contrast, combining data augmentations with ARDS delivers the largest robustness gains, indicating our robust training mixture and augmentation work in synergy to yield stronger debiasing and robustness improvement.

**Effectiveness without Access to Full Training Data.** We further evaluate ARDS in a practical scenario where the entire training corpus is not available at once, and data arrive dynamically over time. To investigate the influence of data volume on building worst-case evaluation subgroups, we randomly sample 10% of the training data as the initially available subset and treat the remaining 90% as newly incoming training data. See Appendix H.9 for more details. As shown in Table 19 of Appendix H.9, even when exposed to only one-tenth of the full corpus during subgroup construction, ARDS* still achieves notable robustness gains, outperforming both LESS and COINCIDE. This demonstrates the potential and applicability of our approach to more dynamic data selection scenarios.

## 6   Conclusions

In this work, we introduce a simple yet effective gradient-free robustness-aware data selection approach for robust visual instruction tuning of large multimodal models. To enhance model robustness against underlying dataset biases, our method first constructs a conversation vector database and then performs deep hierarchical clustering and applies dual perturbations to build worst-case evaluation subgroups. By identifying training data points that are most similar to samples in each subgroup—particularly those susceptible to biased model behavior—we retrieve the highest-scoring instances to curate the robust training mixture. Experimental results demonstrate that performing visual instruction tuning on our mixture with only 30% of original training data achieves a large robustness improvement and comparable clean performance compared to a full-data-trained model. Future work includes broadening the definition of robustness beyond the current bias types and extending data selection for more diverse and dynamic real-world applications.

## Acknowledgement

This work is supported by National Natural Science Foundation of China (NSFC Project No. 62306250) and City University of Hong Kong (CityU Project No. 9610614).

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

# Data Selection Matters: Towards Robust Instruction Tuning of Large Multimodal Models
## *-Supplementary Material-*

# A  Limitations

In this paper, we present a robustness-aware visual instruction selection framework, paving a new path for enhancing the robustness of large multimodal models. Extensive experiments demonstrate that our approach improves robustness to symbol–content spurious correlations, positional bias, and fine-grained visual sub-population shifts across multiple downstream tasks. Although the efficacy of the proposed method has been confirmed, opportunities for refinement persist. One avenue for improvement involves a more diverse investigation into the potential dataset biases and their interaction effects. We also plan to extend the ARDS to other post-training and test-time training scenarios where a few informative training samples are collected to align the model for diverse real-world applications.

# B  Broader Impact

This work presents a novel approach to improving the robustness and data efficiency of visual instruction tuning for large multimodal models, which can have significant implications for reducing biases in AI systems. By mitigating the risks associated with dataset biases, our method could contribute to more fair, transparent, and trustworthy AI applications across a variety of domains.

# C  More Related Work

**Coreset Selection.** Coreset selection attempts to extract a high-quality and most informative subset of training data [84]. Existing coreset selection methods can be mainly categorized into geometry-based [94, 4], uncertainty-based [20, 38], error-based [83, 101], gradient-based [77, 48], and decision boundary-based methods [26, 110]. We refer readers to [35] for a more detailed literature compilation. Our perturbation-based worst-case subgroup construction is closest to [110], which used adversarial gradient ascent to find samples near the decision boundary. However, our gradient-free approach enables efficient identification of vulnerable samples for robustness improvement.

**Active Learning.** Both data selection and active learning aim to identify a subset of data that yields the best possible model, while the distinction lies in the focus on settings. Active learning defaults to that data is unlabled and emphasizes on reducing annotation cost [43, 29, 27, 71, 55]. In contrast, data selection methods usually have full access to all labels when selecting [7], and the goal is to improve training efficiency [107] or robustness [44]. Our method ARDS is a data selection method with no extra annotation required. Our focus is to boost robustness against specific dataset biases for visual instruction tuning while avoiding computation-intensive gradient-based calculation in [107, 44].

**Data Selection for Instruction Tuning.** Recent instruction-following data selection methods show promising results that carefully chosen subsets can align large language models (LLMs) as well as—or better than—using the entire corpus, greatly improving data efficiency [121, 14, 66, 39, 62, 107, 7]. LIMA [121] was the first work to demonstrate that roughly one thousand human-curated examples are sufficient for enabling LLM to follow instructions. Subsequent studies are proposed to use various information measure strategies and emphasize different aspects of data while eliminating human intervention. Some score-based methods directly utilize external closed-source models to score quality and complexity [14, 66] or emphasize diversity [12] when ranking instruction data. Other works introduce natural language indicators [13] or predictive uncertainty [65] for quality estimation, utilize condition-based losses to capture difficulty [57], and compute Shapley value to quantify sample interactions [39]. Reference-based methods require a well-trained model for calculating metrics. For example, RHO-1 [62] utilizes the excess loss [89, 76] to selectively train on useful and learnable tokens. Our setting is more similar to targeted instruction tuning [107], which ranks candidate instructions by their gradient cosine similarity to a few examples from a specific downstream task. Our work extends this line of research to curate robust training mixtures. We select data that mitigates specific dataset bias without any gradient calculations, strengthening the robustness of visual instruction tuning.

# D Detailed Experimental Setup

**Training Datasets.** We curate the robust training mixture by conducting visual instruction selection from the original LLaVA-1.5 dataset, which contains 665K multimodal conversations [63] collected from mixed sources, such as LLaVA-158K [64], VQAv2 [34], OKVQA [73], RefCOCO [46], and TextCaps [97]. These conversations cover a broad spectrum of task types, including visual-question answering, detailed image/region caption, localization and complex visual reasoning. Our mixture is sampled directly from this corpus without altering the original data or introducing external examples. We refer readers to LLaVA [64, 63] for a more detailed dataset description.

**Evaluation Setup.** Following LLaVA-1.5 [63], we assess the zero-shot generalization capability of visual instruction tuning on eight benchmarks including ScienceQA [70], SEED-Bench [52], MMBench [67], MMMU [114], BoolQ [18], ARC [19], A-OKVQA [93] and GQA [42]. For MMBench, we report the accuracy without circular evaluation because we apply stronger adversarial permutation attacks (introduced below). We inject symbol and adversarial permutation attacks as input variations for every evaluation task to probe model robustness against dataset biases, including symbol-content spurious correlation and position bias. The adversarial permutation attack (PA) is incorporated by generating all possible permutations via brute-force algorithms (i.e., $k!$ permutations, where $k$ is the number of options), which is cheaper than gradient-based search at small $k$. The symbol attack (SA) involves a large candidate space and we simply replace canonical choice labels (A/B/C/D) with an alternative set (Q/W/E/R), disrupting symbol–content shortcuts. Robust accuracy is defined as the proportion of tasks that the model answers questions correctly under all perturbations. We additionally evaluate robustness to visual shortcuts by following the GQA-OOD protocol [47], which constructs fine-grained subpopulation shifts for assessment.

**Architecture.** We consider LLaVA-1.5 [63] in our main experiments which consists of a CLIP-ViT-L-336 visual encoder ($336 \times 336$) [87], a two-layer cross-modal MLP projector and Vicuna v1.5 [119] as the language backbone. We initialize the MLP projector with the official LLaVA pre-trained weights and focus on the visual instruction tuning stage. Only the projector and the large language model are updated during visual instruction tuning while keeping the CLIP encoder frozen.

**Implementation Details of Data Selection.** We follow LESS [107] and adopt a warmed-up LLaVA-1.5 (7B) model as our proxy model for data selection. The warm-up stage fine-tunes the model on 1000 randomly sampled training examples for four epochs. With the warmed-up model, we build the attention-weighted conversation-vector database on the training corpus in which each vector is $r \in \mathbb{R}^{2 \times d}$ and the hidden state dimension $d$ of the last embedding layer is 4096. To construct the worst-case evaluation subgroups, we first apply hierarchical clustering to conversation vectors from different multimodal tasks [68] (e.g., image captioning, regional description, open-ended visual question answering). We set the number of clusters $K$ to 70 and the subgroup budget $B$ to 50. To identify vulnerable samples near the decision boundary in subgroups, we inject task-specific perturbations for robustness against targeted dataset biases. Specifically, we impose diffusion noise [122] with the step size of 300 as the visual perturbation and inject symbol and permutation attacks to expose position bias and symbol-content shortcuts. For visual spurious correlation in GQA-OOD, we only incorporate diffusion noise. The robust training mixture is curated by selecting samples that contribute most to the worst-case subgroups. We then perform visual instruction tuning on this mixture using the standard LLaVA pipeline, leaving the original data and training procedure unchanged.

**Implementation Details of Visual Instruction Tuning.** We do not perform hyperparameter sweeps of visual instruction tuning for either our method or the baselines; instead, we adopt the official LLaVA-1.5 configurations [63] so that results under full-data training and data selection baselines are directly comparable. The CLIP image encoder is kept frozen, whereas the language model is updated during supervised visual instruction tuning. We train for one epoch with a batch size of 128 and a learning rate of $2e - 4$ decayed by a cosine schedule. To reduce the memory requirements, we apply Low-Rank Adaptation (LoRA) [41] to all linear layers in the multi-head self-attention (MHSA) and feedforward network (FFN) modules of Vicuna v1.5 [119]. A complete list of hyperparameters is provided in Table 8. At test time we use greedy decoding on every evaluation benchmark to ensure full reproducibility.

**Baselines** We provide a more detailed explanation of the baselines.

- **Random** sampling method randomly selects a subset from the entire training mixture.

Table 8: **Hyperparameters of visual instruction tuning**

(a) LLaVA-1.5 (7B)

| Hyperparameter | Finetune |
|---|---|
| LoRA rank | 128 |
| batch size | 128 |
| lr | 2e-4 |
| lr schedule | cosine decay |
| lr warmup ratio | 0.03 |
| weight decay | 0 |
| epoch | 1 |
| optimizer | AdamW |
| DeepSpeed stage | 3 |

(b) LLaVA-1.5 (13B)

| Hyperparameter | Finetune |
|---|---|
| LoRA rank | 128 |
| batch size | 128 |
| lr | 2e-5 |
| lr schedule | cosine decay |
| lr warmup ratio | 0.03 |
| weight decay | 0 |
| epoch | 1 |
| optimizer | AdamW |
| DeepSpeed stage | 3 |

- **LESS** [107] modified gradient-based influence function that ranks training samples by the cosine similarity between their gradients and those of a few labeled downstream examples for targeted instruction tuning. The downstream examples are also leveraged for test-time in-context learning. For a fairer zero-shot comparison, we only utilize downstream examples for data selection. We use the same warmed-up proxy model for LESS and our methods. To build the gradient store, all Adam-based LoRA gradients are extracted and concatenated for each training sample, which are then reduced to an 8192-dimensional vector using TRAK projection [82]. LESS-SciQA denotes employing 100 examples randomly chosen from the ScienceQA validation split. The training samples with higher gradient cosine similarity to these examples are selected for visual instruction tuning.
- **RHO-LOSS** [76] exemplifies the excess-loss family of selectors [76, 108, 62]. We treat our worst-case evaluation subgroups as a holdout set and train the reference model on those subgroups. Each candidate training sample is then scored by the loss difference between the still-untrained target model and that of the reference model. Samples with larger excess loss, deemed both difficult and learnable, are chosen for visual-instruction tuning.
- **COINCIDE** [51] utilizes a proxy model, well-trained on the entire training corpus, to extract feature embeddings from five multi-head self-attention layers (i.e., 3, 7, 11, 15, 19 layers). The uniform clustering is then performed on these embeddings to form 10000 clusters. COINCIDE selects training samples from each cluster with an emphasis on diversity, drawing more samples from low-density and highly transferable clusters. We reproduce its best version with TinyLLaVA-2B [120] as the proxy model.

## E  Hyper-parameter Studies of ARDS

To better understand the sensitivity and design choices of our proposed ARDS method, we conduct controlled experiments on several key hyperparameters. Since we adopt the same proxy model setup as the baseline LESS [107] for a fair comparison, we report the performance of reimplemented LESS with the selection ratio 5% in Table 9 (a). The results show that training the proxy model for just 4 epochs on a small subset of 1,000 samples is already sufficient for effective data selection. The results in Table 9 (b) demonstrate that doubling the number of clusters from 70 to 140 does not lead to performance improvement, suggesting over-segmentation can dilute subgroup-level signals, reducing the effectiveness of worst-case evaluation subgroup construction. As shown in Table 9 (c), a moderate expansion of the subgroup selection budget helps capture more informative training examples that contribute to robustness.

## F  More Results Across Training Data Scales

To better understand the performance of our ARDS under varying training budgets, we conduct a robustness scaling analysis across different training mixture sizes—5%, 10%, 30%, 50%, and 70% of the full dataset. As shown in Figure 4, we observe that random subsampling fails to improve robustness across all scales. In many cases, it actually leads to performance degradation, especially under stronger perturbation settings. In contrast, both variants of our ARDS consistently outperform baselines at every training size and enhance the robustness of visual instruction tuning, surpassing the

Table 9: Hyper-paraemter studies of ARDS. **(a)** Effect of different warmup configurations for the proxy models. **(b)** Effect of cluster size in ARDS. **(c)** Effect of subgroup budget. $\star$ denotes our choice.

**(a) Warmup Training**

| Epoch | #Sample | ScienceQA |
|---|---|---|
| 4 | 33265 | 67.79 |
| 1 | 1000 | 66.88 |
| 4$\star$ | 1000 | 68.37 |

**(b) Cluster Size**

| Method | ScienceQA | |
|---|---|---|
| | Clean | Robust |
| 70$\star$ | 69.26 | 47.60 |
| 140 | 67.58 | 46.26 |

**(c) Subgroup Budget**

| Method | GQA | |
|---|---|---|
| | Clean | Robust |
| 20 | 61.75 | 51.18 |
| 50$\star$ | 62.43 | 52.21 |

full-data trained model. We choose 30% as the default training budget in our main experiments, as it offers the best trade-off between data efficiency and both clean and robust performance.

# G   More Ablation Studies

**Score aggregation strategy.** To better understand the impact of the score aggregation strategy used in our robustness-aware selection, we conduct an ablation comparing two approaches. The Subgroup Maximum strategy follows the Equation 2 used in targeted instruction tuning [107], where the information score $\mathcal{I}(x_i)$ is computed as the maximum cosine similarity between a training sample and any single worst-case subgroup. The Subgroup Weighted Sum approach considers both subgroup similarity and subgroup difficulty. Specifically, we first compute the cosine similarity $d_{i\mathcal{S}_m}$ between the training sample and each worst-case evaluation subgroup $\mathcal{S}_m$, then weight each similarity by the subgroup's difficulty $\ell_{\mathcal{S}_m}$ using a softmax normalization. This weighting encourages prioritizing training samples that are close to more difficult subgroups, enabling the selection of more informative and robustness-critical examples. As shown in Table 10, the weighted sum strategy outperforms the maximum-based aggregation, yielding higher robust accuracy under more challenging attack settings. This suggests that incorporating subgroup difficulty helps select training samples that more effectively target model-biased behaviors.

Table 10: Ablation study comparing different score aggregation strategies for robust data selection. LLaVA-1.5 (7B) is utilized as the proxy and target model for all methods.

| Score Aggregation Strategy | Data Percentage | ScienceQA | | | | | SEED-Bench | | | | |
|---|---|---|---|---|---|---|---|---|---|---|---|
| | | Clean | PA | SA | SA + PA | Avg. | Clean | PA | SA | SA + PA | Avg. |
| Subgroup Maximum | 30% | 70.05 | 57.61 | 68.12 | 43.88 | 59.91 | 57.23 | 41.02 | 56.13 | 30.65 | 46.25 |
| Subgroup Weighted Sum | 30% | 69.26 | 59.40 | 68.57 | 47.60 | 61.21 | 58.11 | 40.73 | 56.83 | 31.52 | 46.80 |

**Impact of visual and textual perturbations.** We conduct a finer-grained ablation to compare variants of our method using only diffusion-based visual noise or task-aware textual perturbations (PA/SA) against the final dual perturbation strategy. As shown in Table 11, results highlight the individual and combined contributions of visual and textual perturbations in constructing worst-case evaluation subgroups.

Table 11: Ablation study comparing different perturbations for constructing worst-case evaluation subgroups. LLaVA-1.5 (7B) is utilized as the proxy and target model for all methods.

| Textual Perturbation | Visual Perturbation | Data Percentage | ScienceQA | | | | | SEED-Bench | | | | |
|---|---|---|---|---|---|---|---|---|---|---|---|---|
| | | | Clean | PA | SA | SA + PA | Avg. | Clean | PA | SA | SA + PA | Avg. |
| ✓ | ✗ | 10% | 68.82 | 58.25 | 65.74 | 45.61 | 59.61 | 54.08 | 37.82 | 52.60 | 28.45 | 43.23 |
| ✗ | ✓ | 10% | 68.52 | 54.09 | 66.83 | 47.05 | 59.12 | 54.34 | 37.64 | 53.72 | 32.42 | 44.53 |
| ✓ | ✓ | 10% | 69.66 | 55.88 | 69.21 | 52.35 | 61.78 | 53.86 | 38.30 | 53.38 | 35.11 | 45.16 |

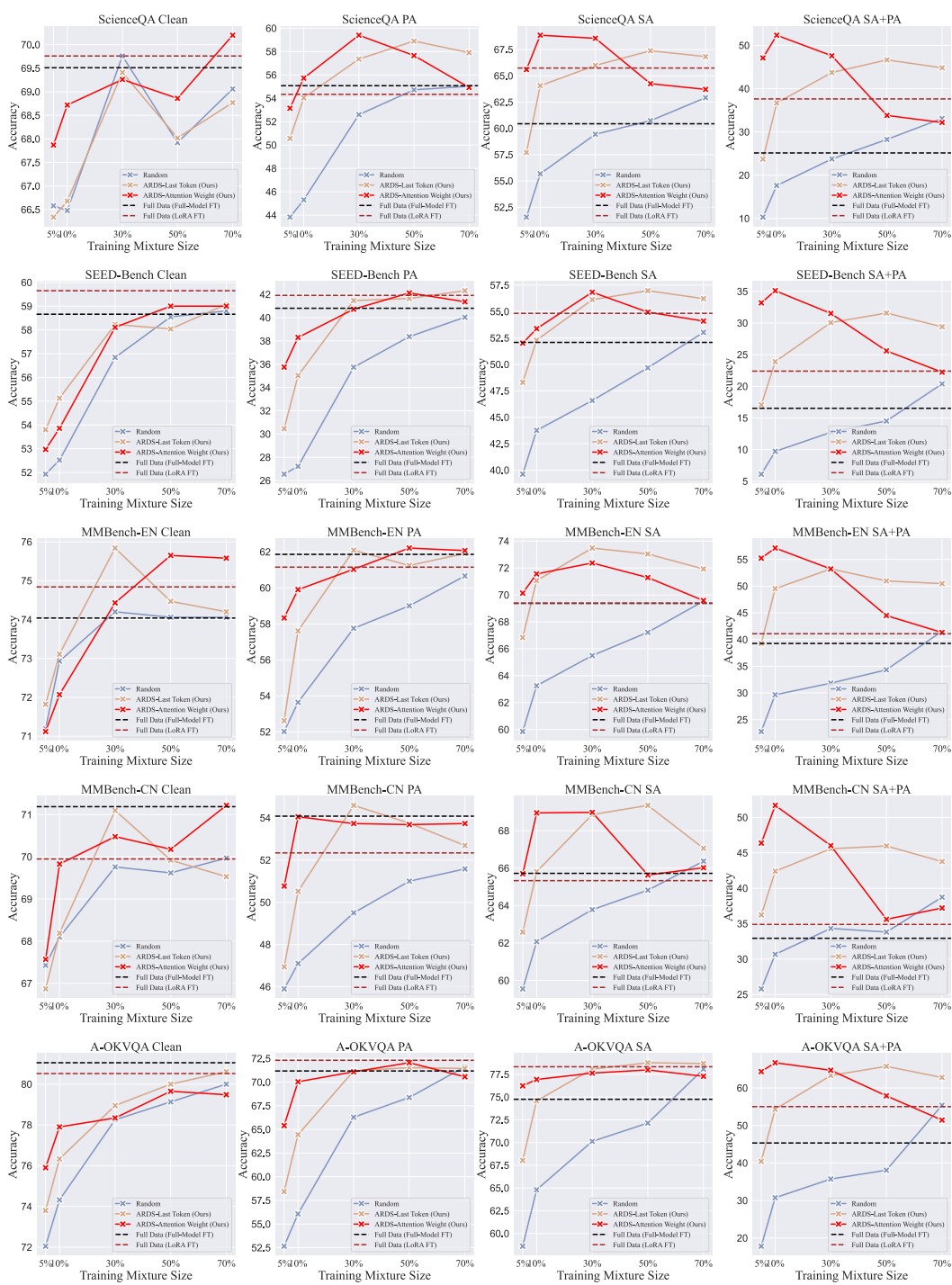

Figure 4: Zero-shot robust accuracy (↑) across varying training mixture sizes for visual instruction tuning of LLaVA-1.5 on the ScienceQA, SEED-Bench, MMBench-EN, MMBench-CN, A-OKVQA benchmarks. Our method consistently improves robustness across scales.

# H More Analayses

## H.1 Comparison Between Full-model and LoRA Fine-tuning

In this section, we compare full-parameter fine-tuning with Low-Rank Adaptation (LoRA) [41]. As reported in Table 12, updating only 4.6% of the parameters via LoRA not only preserves clean-set accuracy but improves robustness. In our main experiments, visual instruction tuning with the full training corpus and every data selection baseline adopts the same LoRA configuration to ensure a fair comparison.

Table 12: Robustness comparison between Full-model and LoRA [41] visual instruction tuning for LLaVA-1.5 (7B) on the full dataset.

| Selection Method | Data Percentage | ScienceQA | | | | | SEED-Bench | | | | | MMBench-EN | | | | | A-OKVQA | | | | |
|---|---|---|---|---|---|---|---|---|---|---|---|---|---|---|---|---|---|---|---|---|---|
| | | Clean | PA | SA | SA + PA | Avg. | Clean | PA | SA | SA + PA | Avg. | Clean | PA | SA | SA + PA | Avg. | Clean | PA | SA | SA + PA | Avg. |
| **Full** | 100% | 69.51 | 55.08 | 60.44 | 25.14 | 52.54 | 58.66 | 40.81 | 52.08 | 16.53 | 42.02 | 74.04 | 61.86 | 69.37 | 39.27 | 61.13 | 81.05 | 71.18 | 74.76 | 45.33 | 68.08 |
| **Full-LoRA** | 100% | 69.76 | 54.34 | 65.74 | 37.63 | 56.87 | 59.65 | 41.92 | 54.83 | 22.40 | 44.69 | 74.84 | 61.15 | 69.39 | 41.09 | 61.62 | 80.52 | 72.31 | 78.34 | 55.02 | 71.55 |

## H.2 Results of Generalization to More Challenging Benchmarks

We further evaluate the generalization capability of our curated robust training mixture on several more challenging benchmarks, including the text-only benchmark, SocialIQA [91], and two math-related visual reasoning benchmarks, MathVista [69] and DynaMath [126]. These two mathematical benchmarks represent more difficult Out-of-Domain (OOD) tasks, as the original LLaVA–665K dataset does not contain explicit mathematical training data [63]. Specifically, MathVista [69] assesses mathematical reasoning in visual complex scenarios (e.g., tables and function plots), while DynaMath [126] is designed to systematically analyze the robustness of mathematical reasoning across diverse topics (e.g., arithmetic, geometry and graph theory) under dynamic condition changes (e.g., numerical value variants, geometric transformations and graph structure variants). Our proposed permutation and symbol attacks, serving as an orthogonal analysis axis, offer a complementary perspective for diagnosing potential biased behaviors in large multimodal models. As shown in Table 13, our curated robust training mixture consistently enhances average performance across OOD tasks for visual instruction tuning, outperforming both full-data training and previous state-of-the-art data selection strategies.

Table 13: **Zero-shot robust accuracies (%, ↑)** against spurious correlation and position bias on additional benchmarks. Results are reported on the text-only SocialIQA and two visual mathematical reasoning benchmarks, MathVista and DynaMath.

| Selection Method | Data Percentage | SocialIQA | | | | | MathVista | | | | | DynaMath | | | | |
|---|---|---|---|---|---|---|---|---|---|---|---|---|---|---|---|---|
| | | Clean | PA | SA | SA + PA | Avg. | Clean | PA | SA | SA + PA | Avg. | Clean | PA | SA | SA + PA | Avg. |
| **Full** | 100% | 66.17 | 51.59 | 55.83 | 21.24 | 48.70 | 40.37 | 16.48 | 34.44 | 2.22 | 23.38 | 39.53 | 19.80 | 36.88 | 1.74 | 24.48 |
| **Random** | 30% | 68.83 | 52.97 | 57.22 | 18.01 | 49.25 | 39.44 | 15.56 | 23.15 | 1.30 | 19.86 | **38.39** | 16.97 | 27.32 | 1.56 | 21.06 |
| **LESS** [107] | 30% | 66.22 | 51.54 | 59.37 | 26.61 | 50.93 | 36.85 | 19.44 | 31.85 | 5.37 | 23.38 | 35.56 | **20.88** | 36.28 | 8.06 | 25.19 |
| **COINCIDE** [51] | 30% | **68.94** | 52.05 | 60.29 | 27.94 | 52.30 | 37.41 | 11.48 | 25.00 | 2.04 | 18.98 | 35.74 | 14.80 | 26.90 | 2.47 | 19.98 |
| **ARDS (ours)** | 30% | 68.63 | **57.63** | **65.10** | **42.89** | **58.56** | 39.81 | **20.93** | 32.78 | **6.48** | **25.00** | 36.40 | 20.22 | **36.76** | **11.19** | **26.14** |

## H.3 Results on Open-Ended Generation Benchmarks

Inspired by LESS [107], ARDS is a targeted data selection method designed to effectively and efficiently perform task-aware instruction tuning on a small selected subset of data once the conversation vector database is established. However, ARDS explicitly aims to enhance robustness against intrinsic dataset biases in visual instruction tuning. In this section, to assess the adaptability of our framework for open-ended generation tasks, we evaluate ARDS on two representative benchmarks, TextVQA [98] and GQA [42]. We denote our robustness-oriented variant in our main experiments as ARDS-Robust and the variant targeting open-ended generation capability as ARDS-OG. As shown in Table 14, our method achieves performance comparable to full-data training and previous data selectors optimized for clean data efficiency. We further observe a natural trade-off between

robustness and generation capability. A concurrent work ICONS [106] extends the line of targeted instruction selection toward multi-task selection via majority voting. We believe such approaches as complementary to our robustness-aware selection and could be integrated into our framework for exploring robustness-aware and multi-objective instruction selection, which we leave as future work.

Table 14: Comparison of data selection methods on open-ended generation capability.

| Selection Method | Data Percentage | TextVQA | GQA |
|---|---|---|---|
| **Full** | 100% | 57.23 | 61.94 |
| **Random** | 30% | 55.90 | 59.61 |
| **LESS-SciQA** [107] | 30% | 49.45 | 56.49 |
| **COINCIDE** [51] | 30% | 56.09 | 59.15 |
| **ARDS-Robust** | 30% | 54.51 | 58.18 |
| **ARDS-OG** | 30% | 56.97 | 60.32 |

## H.4   Generalization to Other Large Multimodal Models

To assess the generality and effectiveness of ARDS, we further conduct a transferability study across a range of backbone architectures beyond the Vicuna-based LLaVA-1.5 used in our main experiments. Specifically, we apply ARDS to two representative state-of-the-art large multimodal models, LLaVA-1.6-Mistral [45] (denoted LLaVA-Mistral) and Qwen2.5-VL-Instruct [10] (denoted Qwen2.5-VL). For LLaVA-Mistral, we follow the official two-stage training procedure of LLaVA-1.5, first pre-training the MLP projector and then performing visual instruction tuning. For Qwen2.5-VL-Instruct, whose pre-training data are closed-source and no publicly released pre-trained version is available, we directly perform post-training. All models are trained on the LLaVA-665K dataset using different data selection strategies. As summarized in Table 15, ARDS consistently enhances robustness and data efficiency across architectures, yielding clear gains in both visual instruction tuning and post-training settings. The results collectively demonstrate the adaptability of ARDS and can be easily applied to various large multimodal models to achieve consistent robustness improvements.

Table 15: **Transferability across large multimodal architectures.** The robust data mixture curated with Vicuna-based LLaVA-1.5 (7B) transfers effectively to other architectures, including LLaVA-1.6-Mistral (7B) and Qwen2.5-VL-Instruct (7B), yielding consistent robustness improvements across visual instruction tuning and post-training settings.

| Proxy Model | Target Model | Selection Method | Data Percentage | ScienceQA Clean | PA | SA | SA + PA | Avg. | SEED-Bench Clean | PA | SA | SA + PA | Avg. | MMBench-EN Clean | PA | SA | SA + PA | Avg. |
|---|---|---|---|---|---|---|---|---|---|---|---|---|---|---|---|---|---|---|
| - | LLaVA-Mistral (7B) | **Full** | 100% | 73.03 | 60.78 | 68.32 | 42.79 | 61.23 | 59.22 | 39.65 | 56.62 | 28.98 | 46.11 | 77.04 | 62.05 | 73.30 | 47.05 | 64.86 |
| - | LLaVA-Mistral (7B) | **Random** | 30% | 73.08 | 56.22 | 58.70 | 21.17 | 52.29 | 56.84 | 34.85 | 50.47 | 14.05 | 39.05 | 75.31 | 58.51 | 67.48 | 32.87 | 58.54 |
| LLaVA-1.5 (7B) | LLaVA-Mistral (7B) | **ARDS** | 30% | 72.04 | 61.77 | 69.16 | 55.53 | **64.63** | 59.22 | 44.02 | 57.53 | 34.93 | **48.93** | 76.97 | 65.37 | 75.17 | 55.19 | **68.18** |
| - | Qwen2.5-VL (7B) | - | - | 77.05 | 63.71 | 67.08 | 33.71 | 60.38 | 48.61 | 24.72 | 53.09 | 10.60 | 34.25 | 71.31 | 52.48 | 72.14 | 35.16 | 57.77 |
| - | Qwen2.5-VL (7B) | **Random** | 30% | 80.32 | 69.31 | 67.43 | 31.78 | 62.21 | 52.06 | 28.50 | 53.67 | 8.98 | 35.80 | 74.27 | 57.36 | 73.83 | 34.63 | 60.02 |
| LLaVA-1.5 (7B) | Qwen2.5-VL (7B) | **ARDS** | 30% | 83.84 | 76.55 | 70.15 | 36.19 | **66.68** | 61.71 | 41.81 | 55.40 | 10.46 | **42.35** | 80.85 | 69.81 | 75.44 | 40.29 | **66.60** |

| Proxy Model | Target Model | Selection Method | Data Percentage | MMBench-CN Clean | PA | SA | SA + PA | Avg. | A-OKVQA Clean | PA | SA | SA + PA | Avg. | MMMU Clean | PA | SA | SA + PA | Avg. |
|---|---|---|---|---|---|---|---|---|---|---|---|---|---|---|---|---|---|---|
| - | LLaVA-Mistral (7B) | Full | 100% | 71.63 | 52.34 | 66.99 | 38.51 | 57.36 | 80.00 | 68.38 | 77.99 | 59.21 | 71.39 | 38.84 | 12.51 | 35.54 | 6.49 | 23.34 |
| - | LLaVA-Mistral (7B) | Random | 30% | 68.33 | 49.04 | 57.57 | 13.17 | 47.02 | 77.47 | 61.31 | 72.93 | 39.21 | 62.73 | 37.43 | 12.51 | 35.30 | 3.07 | 22.07 |
| LLaVA-1.5 (7B) | LLaVA-Mistral (7B) | **ARDS** | 30% | 72.26 | 57.84 | 70.32 | 51.24 | **62.92** | 81.66 | 72.58 | 80.52 | 69.00 | **75.94** | 39.55 | 16.06 | 36.60 | 11.33 | **25.89** |
| - | Qwen2.5-VL (7B) | - | - | 63.62 | 36.59 | 73.60 | 36.71 | 52.63 | 82.18 | 67.34 | 75.90 | 41.48 | 66.72 | 52.66 | 26.21 | 45.45 | 11.92 | 34.06 |
| - | Qwen2.5-VL (7B) | Random | 30% | 68.05 | 41.79 | 73.46 | 32.92 | 54.05 | 84.54 | 73.01 | 75.90 | 38.25 | 67.92 | 53.72 | 26.56 | 46.40 | 10.74 | 34.35 |
| LLaVA-1.5 (7B) | Qwen2.5-VL (7B) | **ARDS** | 30% | 79.05 | 63.99 | 75.88 | 38.58 | **64.38** | 85.85 | 77.03 | 77.55 | 42.01 | **70.61** | 53.13 | 26.92 | 45.93 | 11.57 | **34.39** |

## H.5   Statistical Analysis of the Loss Distribution in Worst-case Evaluation Subgroups

To analyze the loss distribution of worst-case evaluation subgroups, we measure balance through Shannon entropy given as $e(\ell_{\mathcal{S}}) = -\sum_{m=0}^{M} p(\ell_{\mathcal{S}_m}) \log p(\ell_{\mathcal{S}_m})$, where $p(\ell_{\mathcal{S}_m})$ is the weight of the $m$-th cluster after $\mathrm{Softmax}(\ell_{\mathcal{S}_m})$. As entropy depends on the number of subgroups, we further normalize entropy w.r.t. the number $M$ of possible clusters: $\bar{e}(\ell_{\mathcal{S}}) = \frac{e(\ell_{\mathcal{S}})}{\log(M)}$, where $\log(M)$ is equal to the entropy of a uniform distribution of size $M$. We obtained the normalized entropy $\bar{e}(\ell_{\mathcal{S}}) = 0.92$. The result empirically verifies that selected samples remain well dispersed under the loss-based weighted-sum aggregation strategy. Furthermore, as shown in Table 10, the weighted-sum

aggregation strategy yields a better clean and robustness trade-off and avoids the skew that the max variant exhibits.

## H.6 Generalization under Distribution Shifts and Visual Corruptions

To further evaluate the robustness and generalization capability of our robust training mixture, we conduct experiments on previously unseen perturbations not used during data selection, including unseen symbol attacks and stronger visual corruptions. Specifically, the canonical answer labels A/B/C/D are replaced by S/N/V/F and U/I/O/P, respectively, and the stronger diffusion noise is injected into the test images. The adversarial permutation attack is adaptively generated for each new question by enumerating all possible permutations of the answer options. As shown in Table 17 and Table 16, our ARDS maintains the highest robust accuracy across all scenarios, demonstrating that ARDS effectively prioritizes training examples that confer transferable robustness.

Table 16: Transferable robustness under image corruptions. **SA**: symbol attack, **PA**: permutation attack. **IC**: image corruption.

| Selection Method | Data Percentage | ScienceQA | | | | | | | | | |
|---|---|---|---|---|---|---|---|---|---|---|---|
| | | Clean | PA | SA | SA+PA | Avg. | IC | IC+PA | IC+SA | IC+SA+PA | Avg. |
| **Full** | 100% | 69.76 | 54.34 | 65.74 | 37.63 | 56.87 | 64.75 | 47.84 | 61.33 | 31.38 | 51.32 |
| **Random** | 30% | 69.76 | 52.60 | 59.44 | 23.75 | 51.39 | 67.43 | 47.60 | 55.83 | 20.13 | 47.74 |
| **ARDS (Ours)** | 30% | 69.26 | 59.40 | 68.57 | 47.60 | 61.21 | 66.04 | 53.94 | 64.80 | 41.55 | 56.58 |

Table 17: **Transferable robustness under unseen attacks**. **PA**: permutation attack, **SA1**: unseen symbol attack (S/N/V/F), **SA2**: unseen symbol attack (U/I/O/P). **WCS**: incorporate our worst-case evaluation subgroups during data selection.

| Selection Method | Data Percentage | ScienceQA | | | | | SEED–Bench | | | | | A-OKVQA | | | | | MMBench–EN | | | | |
|---|---|---|---|---|---|---|---|---|---|---|---|---|---|---|---|---|---|---|---|---|---|
| | | SA1 | SA1 + PA | SA2 | SA2 + PA | Avg. | SA1 | SA1 + PA | SA2 | SA2 + PA | Avg. | SA1 | SA1 + PA | SA2 | SA2 + PA | Avg. | SA1 | SA1 + PA | SA2 | SA2 + PA | Avg. |
| Full | 100% | 67.72 | 40.80 | 66.73 | 36.49 | 52.93 | 56.62 | 25.47 | 55.04 | 19.88 | 39.25 | 77.21 | 52.40 | 76.07 | 48.03 | 63.42 | 69.05 | 46.29 | 68.91 | 36.61 | 55.21 |
| Random | 30% | 64.06 | 32.92 | 63.81 | 31.28 | 48.01 | 47.82 | 13.57 | 47.22 | 8.29 | 29.22 | 60.26 | 24.98 | 65.15 | 18.34 | 42.18 | 60.18 | 29.36 | 62.51 | 21.30 | 43.33 |
| COINCIDE [51] | 30% | 65.44 | 37.04 | 63.41 | 29.85 | 48.94 | 51.65 | 19.40 | 48.58 | 11.82 | 32.86 | 68.47 | 37.03 | 66.55 | 25.68 | 49.43 | 64.22 | 35.27 | 64.29 | 23.98 | 46.94 |
| COINCIDE–WCS | 30% | 62.87 | 33.96 | 63.16 | 31.04 | 47.75 | 50.28 | 16.19 | 51.08 | 16.03 | 33.39 | 67.34 | 29.26 | 71.79 | 35.90 | 51.07 | 63.76 | 30.49 | 65.90 | 26.31 | 46.62 |
| ARDS (ours) | 30% | 68.57 | 48.34 | 66.14 | 39.76 | 55.70 | 57.48 | 33.81 | 54.95 | 22.27 | 42.13 | 78.60 | 63.67 | 74.50 | 49.00 | 66.44 | 71.75 | 51.98 | 70.48 | 42.50 | 59.18 |

## H.7 Complementary with Training Data Augmentation

We take a further step to investigate the impact of simple training data augmentation on mitigating dataset biases. To reduce symbol-content spurious correlation and position biases, we apply random symbol replacement and option shuffling as textual-level augmentations. To address visual spurious correlations and subpopulation shifts, we adopt AutoAugment [22] for image-level augmentation during training. As illustrated in Table 18, while textual augmentations help improve robustness against seen perturbations, they fail to generalize to unseen attacks. This underscores that simple augmentations alone are insufficient to resolve dataset biases. Our approach aims to curate a robust training mixture as a complementary perspective to augmentation. Combining our ARDS with data augmentation results in greater robustness gains, demonstrating their synergistic effect in debiasing model behavior. More critically, AutoAugment does not significantly improve performance on shifted subgroups, and in fact, harms clean accuracy on GQA and robustness on GQA-OOD. This aligns with findings from recent studies, which show that simple data augmentations can generate problematic data and inadvertently amplify existing dataset biases rather than mitigate them, since augmentations typically preserve the statistical properties of the original training data [11, 95, 79].

Table 18: Performance comparisons of different selection methods under the data augmentation during visual instruction tuning. The proxy model and target models are LLaVA-1.5 (7B). **SA**: symbol attack, **PA**: permutation attack, **SA1**: unseen symbol attack (S/N/V/F), **SA2**: unseen symbol attack (U/I/O/P).

| Selection Method | Training Aug. | Data Pct. (Sci.) | Data Pct. (GQA) | ScienceQA | | | | | | | | | | GQA | | | | |
|---|---|---|---|---|---|---|---|---|---|---|---|---|---|---|---|---|---|---|
| | | | | Clean | PA | SA | SA + PA | Avg. | SA1 | SA1 + PA | SA2 | SA2 + PA | Avg. | Clean | OOD-All | OOD-Head | OOD-Tail | Avg. |
| **Full** | ✗ | 100% | 100% | 69.76 | 54.34 | 65.74 | 37.63 | 56.87 | 67.72 | 40.80 | 66.73 | 36.49 | 52.93 | 61.94 | 57.51 | 61.17 | 51.55 | 58.04 |
| | ✓ | 100% | 100% | 68.86 | 57.02 | 70.05 | 54.98 | 62.72 | 69.36 | 49.88 | 68.47 | 42.74 | 57.61 | 59.51 | 55.01 | 58.34 | 49.58 | 55.61 |
| **Random** | ✗ | 30% | 50% | 69.76 | 52.60 | 59.44 | 23.75 | 51.39 | 64.06 | 32.92 | 63.81 | 31.28 | 48.01 | 60.69 | 55.97 | 60.30 | 48.92 | 56.47 |
| | ✓ | 30% | 50% | 68.91 | 52.65 | 69.41 | 49.43 | 60.10 | 66.48 | 42.19 | 64.80 | 34.51 | 51.99 | 56.46 | 51.07 | 54.41 | 45.63 | 51.89 |
| **ARDS (Ours)** | ✗ | 30% | 50% | 69.26 | 59.40 | 68.57 | 47.60 | 61.21 | 68.57 | 48.34 | 66.14 | 39.76 | 55.70 | 62.43 | 58.44 | 62.26 | 52.21 | 58.84 |
| | ✓ | 30% | 50% | 70.40 | 58.90 | 69.21 | 60.14 | 64.66 | 70.30 | 57.06 | 69.11 | 53.64 | 62.53 | 59.64 | 55.22 | 58.45 | 49.95 | 55.82 |

## H.8   Analysis of Performance Degradation on Text-only Tasks

We conduct a detailed analysis of the observed performance degradation of the full-data trained model on pure-text tasks such as ARC-e and BoolQ. We hypothesize that this phenomenon stems from the catastrophic forgetting caused by modality imbalance in large-scale visual instruction tuning, which aligns with prior work that identified text-only forgetting during multimodal alignment [116, 60]. For example, WINGS [116] reports up to 13.33% text-only degradation after visual modality expansion. In LLaVA-665K, only around $40688/665298 \approx 6\%$ of samples are pure-text instructions. During large-scale multimodal tuning, the optimizer increasingly focuses on vision-centric tasks and gradually forgets language knowledge acquired from text-only instruction tuning. To quantify this effect, we measure text-only forgetting by training on random subsets of increasing size using LLaVA-1.5 (7B), as summarized in Figure 5. Larger subsets are expected to include more image-text pairs and thus amplify modality mismatch between multimodal and text-only inputs. The results confirm that incorporating more vision-heavy data helps models excel in multimodal tasks while accentuates forgetting on text-only tasks.

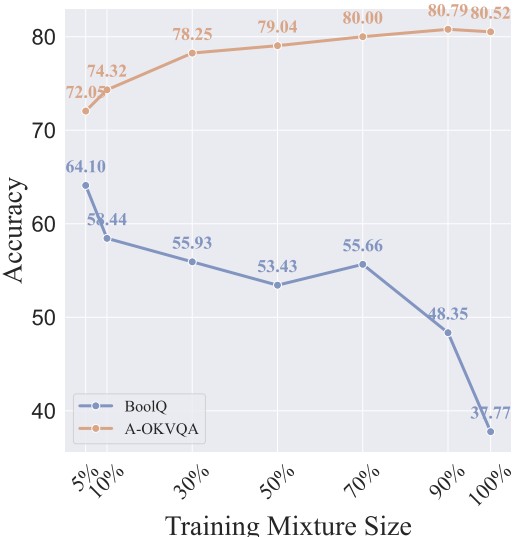

Figure 5: Comparison of multimodal and text-only task performance under different training data scales. As the training subset grows and includes more vision-heavy data, multimodal performance improves while text-only performance degrades, indicating modality-driven forgetting.

## H.9   More Results in Settings without Access to Full Training Data

In this section, we further evaluate ARDS in a practical scenario where the entire training corpus is not available at once, and data arrive dynamically over time. This setting differs from the static setup used in our main experiments, where data selectors have full access to the entire training set to construct gradient databases [107] or perform global clustering [51]. To investigate the influence of data volume on building worst-case evaluation subgroups, we randomly sample 10% of the training data as the initially available subset and treat the remaining 90% as newly incoming training data. We denote the variant as ARDS*. Specifically, ARDS* performs clustering on the 10% subset using the same number of clusters $K$ and subgroup budget $B$, followed by the same task-specific perturbations as the ARDS. As shown in Table 19, even when exposed to only one-tenth of the full corpus during subgroup construction, ARDS* still achieves notable robustness gains, outperforming both LESS and COINCIDE. This demonstrates the potential and applicability of our approach to more dynamic data selection scenarios.

Table 19: **Comparison results in dynamic data availability settings.** ARDS*, which constructs worst-case evaluation subgroups with access to only 10% of the training data, remains effective in improving robustness and outperforms LESS and COINCIDE, the latter of which performs clustering over the full dataset.

| Selection Method | Selection Ratio | ScienceQA | | | | |
|---|---|---|---|---|---|---|
| | | Clean | PA | SA | SA + PA | Avg. |
| **LESS** [107] | 30% | 68.42 | 55.63 | 64.70 | 34.95 | 55.93 |
| **COINCIDE** [51] | 30% | 67.72 | 52.21 | 61.08 | 28.06 | 52.27 |
| **ARDS (ours)** | 30% | 69.26 | 59.40 | 68.57 | 47.60 | 61.21 |
| **ARDS* (ours)** | 30% | 68.86 | 58.30 | 67.13 | 42.39 | 59.17 |

## H.10 More Results on Other Training Datasets

We further evaluate the proposed ARDS on another large-scale instruction-tuning dataset, Vision-Flan [109], to assess its generalizability across different training sources. As shown in Table 20, the robust training mixture curated by ARDS on Vision-Flan [109] achieves superior worst-case robustness compared with full-data and random selection baselines. The results further validate the effectiveness of our robustness-aware data selection framework in mitigating the model's biased behaviors due to spurious correlations and position biases across datasets.

Table 20: **Zero-shot robust accuracies (%, ↑)** against spurious correlation and position bias on Vision-Flan training dataset.

| Selection Method | Selection Ratio | ScienceQA | | | | |
|---|---|---|---|---|---|---|
| | | Clean | PA | SA | SA + PA | Avg. |
| **Full** [107] | 100% | 64.06 | 39.71 | 53.59 | 13.73 | 42.77 |
| **Random** [51] | 50% | 61.38 | 34.41 | 53.59 | 18.89 | 42.06 |
| **ARDS (ours)** | 50% | 62.87 | 37.98 | 60.54 | 33.56 | 48.74 |

## H.11 Robustness Evaluation of Large Multimodal Models

In this section, we provide a more detailed robustness analysis for the large multimodal model trained with standard visual instruction tuning. We perform a series of controlled evaluations by applying textual and visual variations on the GQA benchmark [42]. To study whether the large multimodal models learns instruction-following capability or simply memorize the specific instruction format, we introduce InstructionPerturb by prompting ChatGPT to generate 20 alternative question instructions, such as word replacement, symbol injection, and paraphrasing. Importantly, all variants preserve the original semantic meaning. We then compute the average accuracy across these variants to evaluate how sensitive the model is to instruction format changes. For visual input perturbation, we apply a range of standard test-time augmentation techniques, including color jitter, brightness, sharpness, and spatial transformations (e.g., rotate, shear, translate). Additionally, we evaluate the impact of diffusion noise, which has been shown to more flexibly induce model output incorrect predictions [124, 122]. We report the average robustness performance over diffusion steps 100, 200, and 300. As shown in Figure 6, the model demonstrates moderate robustness to most common textual and visual variations. However, the model exhibits the most significant performance degradation under spurious subpopulation shifts (OOD-Tail [47]). Also, the diffusion noise introduces subtle yet semantically preserving perturbations that can effectively disrupt model predictions, and its strength can be flexibly controlled through the number of diffusion steps. These findings motivate our design of robustness-oriented data selection and evaluation methods.

# I  The ARDS Algorithm

In Algorithm 1, we outline our robustness-aware data selection procedure to curate the robust training mixture.

# J  Theoretical Analysis

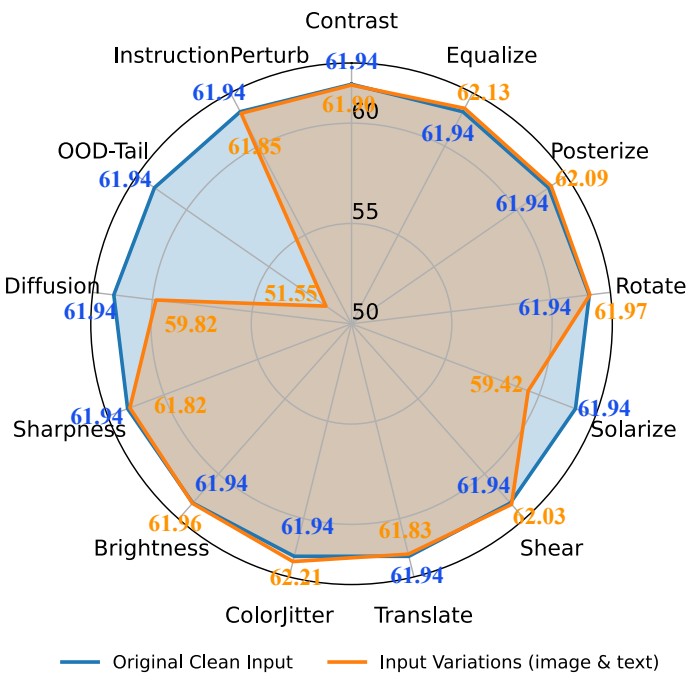

Figure 6: Robustness of LLaVA-1.5 (7B) on the GQA benchmark under diverse input variations.

## J.1 Analysis on the Relationship Between Data Vulnerability and Gradient Magnitude

We begin by exploring the relationship between data vulnerability and gradient magnitude. Intuitively, training samples that are more vulnerable tend to exhibit higher gradient norms, as their losses are more sensitive to parameter updates. This motivates us to identify such samples through worst-case subgroup construction.

We use linear models as examples to demonstrate the theoretical insights of our work. We consider a linear classifier $r(\boldsymbol{x}) = \mathbf{W}\boldsymbol{x}$ where $\boldsymbol{x} \in \mathbb{R}^{d_{in}}$ is the input and $\mathbf{W} \in \mathbb{R}^{d_{out} \times d_{in}}$ represents the parameters. As a classification problem, we use softmax-cross-entropy as the loss objective function: $\mathcal{L}(\boldsymbol{x}, \boldsymbol{y}) = -\boldsymbol{y}^T \log \widehat{\boldsymbol{y}}$ where $\widehat{\boldsymbol{y}} = \text{softmax}(\mathbf{W}\boldsymbol{x})$ is the model's probabilistic output.

The gradient of $\mathcal{L}$ with respect to the model parameters is calculated by:

$$g(\boldsymbol{x}, \boldsymbol{y}) \stackrel{\text{def}}{=} \frac{\partial \mathcal{L}}{\partial \mathbf{W}} = (\widehat{\boldsymbol{y}} - \boldsymbol{y})\boldsymbol{x}^\top \tag{7}$$

The gradient of $\mathcal{L}$ with respect to sample $\boldsymbol{x}$ is calculated by:

$$s(\boldsymbol{x}, \boldsymbol{y}) = \frac{\partial \mathcal{L}}{\partial \boldsymbol{x}} = \mathbf{W}^\top(\widehat{\boldsymbol{y}} - \boldsymbol{y}) \tag{8}$$

**Assumption J.1.** The input data $\boldsymbol{x}$ is normalized, we have $\|\boldsymbol{x}\|_2 = 1$ without the loss of generality.

Assumption J.1 is a benign assumption, as data normalization is quite common in practice.

**Theorem J.2.** *Under Assumption J.1 and in binary classification, i.e., $d_{out} = 2$, if a data instance is adversarially perturbed by PGD, i.e., $\delta = \epsilon \cdot \frac{\partial \mathcal{L}}{\partial \boldsymbol{x}}$, then the data vulnerability, defined by $\mathcal{L}(\boldsymbol{x} + \delta, \boldsymbol{y}) - \mathcal{L}(\boldsymbol{x}, \boldsymbol{y})$, increases monotonically with the gradient magnitude $g(\boldsymbol{x}, \boldsymbol{y})$.*

*Proof.* Without the loss of generality, we assume the data $(\boldsymbol{x}, \boldsymbol{y})$ belongs to the first category. Based on the convexity of the loss function $\mathcal{L}$, we have $\mathcal{L}(\boldsymbol{x} + \delta, \boldsymbol{y}) - \mathcal{L}(\boldsymbol{x}, \boldsymbol{y}) \geq \delta^T \frac{\mathcal{L}(\boldsymbol{x}, \boldsymbol{y})}{\partial \boldsymbol{x}} = \epsilon \|s(\boldsymbol{x}, \boldsymbol{y})\|_2^2$.

---

**Algorithm 1** Data Selection for Robust Visual Instruction Tuning (ARDS)

---

1: **Require:** Training corpus $\mathcal{D}$ of size $N$; Proxy model $f_\theta$; Number of subgroups $K$; Subgroup budget $B$

2: **Step 1: Conversation Vector Database**
3: **for** $i = 1$ to $N$ **do**
4:     Obtain token embeddings $\mathbf{H}_t$ using $f_\theta$ at the last layer with the attention-score matrix $\mathbf{A}$
5:     Aggregate the visual and textual tokens preceding the last token using an attention-score weighted mechanism. $\widehat{\mathbf{H}} = \sum_{t=1}^{L-1} \mathbf{A}_{L,t} \cdot \mathbf{H}_t$
6:     Obtain the conversation vector $r_i = [\mathbf{H}_L; \widehat{\mathbf{H}}]$
7: **end for**

8: **Step 2: Worst-case Evaluation Subgroups** $\mathcal{S}$
9: Perform hierarchical clustering in the embedding space to group a small holdout set sampled from $\mathcal{D}_{\text{train}}$ into $M$ subgroups $\mathcal{C}_m$.
10: Apply perturbations to calculate the loss difference $\mathcal{S}_m = \text{top}_B \{\mathbf{x} \in \mathcal{C}_m : |\ell(\mathbf{x}) - \ell(\mathbf{x}')|\}$
11: For each subgroup, sample up to $\min(|C_k|, B)$ samples most vulnerable to model biased behavior with the highest loss difference to build the worst-case evaluation subgroups.

12: **Step 3: Quality Score Measure and Robust Training Mixture**
13: **for** $i = 1$ to $N$ **do**
14:     Calculate cosine similarity $d_{i\mathcal{S}_m}$ between the conversation vector $r_i$ and each worst-case subgroup $\mathcal{S}_m$
15:     Calculate the quality score $\mathcal{I}(x_i)$ using softmax-weighted aggregation via Eq. (6).
16: **end for**
17: Select training samples with the highest quality scores to form $\mathcal{D}_{\text{robust}}$ for fine-tuning.
18: **Output:** Robust training mixture $\mathcal{D}_{\text{robust}}$

---

On one hand, based on Equation (8), $d_{out} = 2$, $\widehat{\boldsymbol{y}}[1] + \widehat{\boldsymbol{y}}[2] = 1$ and that the data belongs to the first category, we have the following:

$$s(\boldsymbol{x}, \boldsymbol{y}) = \left[\mathbf{W}[1,:]^T, \mathbf{W}[2,:]^T\right] \begin{bmatrix} -\widehat{\boldsymbol{y}}[2] \\ \widehat{\boldsymbol{y}}[2] \end{bmatrix} = \widehat{\boldsymbol{y}}[2] \cdot (\mathbf{W}[2,:]^T - \mathbf{W}[1,:]^T) \tag{9}$$

On the other hand, based on Equation (7), $d_{out} = 2$, $\widehat{\boldsymbol{y}}[1] + \widehat{\boldsymbol{y}}[2] = 1$ and that the data belongs to the first category, we have the following:

$$g(\boldsymbol{x}, \boldsymbol{y}) = \begin{bmatrix} -\widehat{\boldsymbol{y}}[2]\boldsymbol{x}^T \\ \widehat{\boldsymbol{y}}[2]\boldsymbol{x}^T \end{bmatrix} \tag{10}$$

In summary, the vulnerability is lower bounded by $\epsilon\|s(\boldsymbol{x}, \boldsymbol{y})\|_2^2 = \epsilon(\widehat{\boldsymbol{y}}[2])^2\|\mathbf{W}[2,:] - \mathbf{W}[1,:]\|_2^2$, the gradient norm is $\widehat{\boldsymbol{y}}[2] \cdot \|\boldsymbol{x}\|_2$. Considering $\|\boldsymbol{x}\|_2 = 1$ as in Assumption J.1 and the parameter $\mathbf{W}$ is fixed for different data, we can conclude that more vulnerability the data is, the larger magnitude its gradient with respect to the parameters is. $\square$

Theorem J.2 discusses the binary classification case, but each step of a language model is a multi-class classification with large number of categories, which is equal to the vocabulary size. We need to extend the analysis to multi-class cases and start with the following assumption.

**Assumption J.3.** Different rows of the parameter $\mathbf{W}$ have approximately the same magnitude and have low correlation, i.e., $\forall i, j$, we have $\frac{\|\mathbf{W}[i,:]\|_2 - \|\mathbf{W}[j,:]\|_2}{\|\mathbf{W}[j,:]\|_2} = o(1)$ and $\frac{\mathbf{W}[i,:]^T\mathbf{W}[j,:]}{\|\mathbf{W}[i,:]\|_2\|\mathbf{W}[j,:]\|_2} = o(1)$.

Assumption J.3 is a benign assumption, the categories in the classification problem are usually distinct, so the corresponding output vectors usually have small correlation. Assumption J.3 is also consistent with the insights of normalization layers, which is popular in deep learning.

**Theorem J.4.** *Under Assumption J.1 and J.3, if a data instance is adversarially perturbed by PGD, i.e., $\delta = \epsilon \cdot \frac{\partial \mathcal{L}}{\partial \boldsymbol{x}}$, then the data vulnerability, defined by $\mathcal{L}(\boldsymbol{x} + \delta, \boldsymbol{y}) - \mathcal{L}(\boldsymbol{x}, \boldsymbol{y})$, increases monotonically with the gradient magnitude $g(\boldsymbol{x}, \boldsymbol{y})$.*

*Proof.* We follow the technique in the proof of Theorem J.2 and have the bound of the vulnerability $\mathcal{L}(\boldsymbol{x}+\delta,\boldsymbol{y}) - \mathcal{L}(\boldsymbol{x},\boldsymbol{y}) \geq \epsilon\|s(\boldsymbol{x},\boldsymbol{y})\|_2^2$. Without the loss of generality, we assume the data instance belongs to the first category.

Following similar calculations as in Equation (9) and Equation (10), we have the following expression for $s(\boldsymbol{x},\boldsymbol{y})$ and $g(\boldsymbol{x},\boldsymbol{y})$.

$$
s(\boldsymbol{x},\boldsymbol{y}) = \left[\mathbf{W}[1,:]^T, \mathbf{W}[2,:]^T, ..., \mathbf{W}[d_{out},:]^T\right]
\begin{bmatrix}
-\sum_{i=2}^{d_{out}} \widehat{\boldsymbol{y}}[i] \\
\widehat{\boldsymbol{y}}[2] \\
... \\
\widehat{\boldsymbol{y}}[d_{out}]
\end{bmatrix}
= \sum_{i=2}^{d_{out}} \widehat{\boldsymbol{y}}[i] \cdot (\mathbf{W}[i,:]^T - \mathbf{W}[1,:]^T)
$$

(11)

$$
g(\boldsymbol{x},\boldsymbol{y}) =
\begin{bmatrix}
-\left(\sum_{i=2}^{d_{out}} \widehat{\boldsymbol{y}}[i]\right) \boldsymbol{x}^T \\
\widehat{\boldsymbol{y}}[2]\boldsymbol{x}^T \\
... \\
\widehat{\boldsymbol{y}}[d_{out}]\boldsymbol{x}^T
\end{bmatrix}
$$

(12)

Based on Assumption J.3 which means $\mathbf{W}[i,:]$ has approximately the same magnitude and low correlation for different rows, we have:

$$
\|s(\boldsymbol{x},\boldsymbol{y})\|_2^2 = \left\|\sum_{i=2}^{d_{out}} \widehat{\boldsymbol{y}}[i]\mathbf{W}[i,:] - \sum_{i=2}^{d_{out}} \widehat{\boldsymbol{y}}[i]\mathbf{W}[1,:]\right\|_2^2
$$

$$
\simeq \left(\left(\sum_{i=2}^{d_{out}} \widehat{\boldsymbol{y}}[i]\right)^2 + \sum_{i=2}^{d_{out}} \widehat{\boldsymbol{y}}[i]^2\right)\left(\frac{1}{d_{out}}\sum_{i=1}^{d_{out}} \|\mathbf{W}[i,:]\|_2^2\right)
$$

(13)

$$
\|g(\boldsymbol{x},\boldsymbol{y})\|_2^2 = \left(\left(\sum_{i=2}^{d_{out}} \widehat{\boldsymbol{y}}[i]\right)^2 + \sum_{i=2}^{d_{out}} \widehat{\boldsymbol{y}}[i]^2\right)\|\boldsymbol{x}\|_2^2
$$

(14)

Considering $\|\boldsymbol{x}\|_2 = 1$ as in Assumption J.1 and the parameter $\mathbf{W}$ is fixed for different data, we can conclude that more vulnerability the data is, the larger magnitude its gradient with respect to the parameters is. $\square$

## J.2  Analysis on Targeted Instruction Selection for Robustness Improvement

Built on TracIn [85], LESS [107] quantifies the influence of a training datapoint $\boldsymbol{x}$ on the loss of a test data $\boldsymbol{x}'$ via $\mathcal{L}\left(\boldsymbol{x}';\boldsymbol{\theta}^{t+1}\right) - \mathcal{L}\left(\boldsymbol{x}';\boldsymbol{\theta}^t\right) \approx -\eta_t \left\langle \nabla\mathcal{L}\left(\boldsymbol{x};\boldsymbol{\theta}^t\right), \nabla\mathcal{L}\left(\boldsymbol{x}';\boldsymbol{\theta}^t\right)\right\rangle$. Let $\boldsymbol{m}$ denote vulnerable samples selected by our worst-case evaluation subgroups and $\boldsymbol{m}'$ denote a perturbed test sample for robustness evaluation. If the gradient inner-products $\left\langle \nabla\mathcal{L}\left(\boldsymbol{m};\boldsymbol{\theta}^t\right), \nabla\mathcal{L}\left(\boldsymbol{m}';\boldsymbol{\theta}^t\right)\right\rangle$ is notably non-negative, then the loss of perturbed test samples $\mathcal{L}\left(\boldsymbol{m}';\boldsymbol{\theta}^{t+1}\right) - \mathcal{L}\left(\boldsymbol{m}';\boldsymbol{\theta}^t\right)$ decreases. The decreased loss mirrors adversarial training for enhancing adversarial robustness [72]. For empirical validation, we compute this gradient inner product using 100 training batches with the batch size of 32 and 128 perturbed ScienceQA samples. The results of training batches from our robust training mixture show the average $\left\langle \nabla\mathcal{L}\left(\boldsymbol{m};\boldsymbol{\theta}^t\right), \nabla\mathcal{L}\left(\boldsymbol{m}';\boldsymbol{\theta}^t\right)\right\rangle = 25.76$, while that of training batches from random selection is 1.95. This strongly supports our theoretical premise.

## J.3  Analysis on Alignment Between ARDS and Targeted Instruction Selection

To avoid the heavy cost of gradient-based calculations and sidestep the need for access to downstream data, we leverage the rich hidden representation of LMMs and construct worst-case evaluation subgroups on the training corpus. Now we study the cosine similarity discrepancy between feature embeddings and gradients of two data instances. The theorem below demonstrates that the discrepancy is upper bounded, especially for confident correct instances which are quite common in the realm of large language models.

**Theorem J.5.** *Consider two data instances $(\boldsymbol{x}_i, \boldsymbol{y}_i)$ and $(\boldsymbol{x}_j, \boldsymbol{y}_j)$. Let $r_i = r(\boldsymbol{x}_i)$, $g_i = g(\boldsymbol{x}_i, \boldsymbol{y}_i)$ for brevity and we define the corresponding $r_j$, $g_j$. Suppose the maximum and minimum singular values of parameters are $\sigma_1$ and $\sigma_2$. The cosine similarity discrepancy between two feature embeddings and gradients of the $i$-th and $j$-th samples is bounded by*

$$|\Delta_{ij}| \leq 2\frac{\sigma_1^2 + \epsilon^2}{\sigma_2^2} \tag{15}$$

*where $\epsilon = \sup \|\widehat{\boldsymbol{y}} - \boldsymbol{y}\|_2$ is the upper bound of the probability mismatch.*

*Proof.* We begin by explicitly writing $\Delta_{ij}$:

$$
\begin{aligned}
|\Delta_{ij}| &= |\cos(r_i, r_j) - \cos(g_i, g_j)| \\
&= \left| \frac{\langle r_i, r_j \rangle}{\|r_i\|\|r_j\|} - \frac{\langle g_i, g_j \rangle}{\|g_i\|\|g_j\|} \right| \\
&\leq \frac{|\langle r_i, r_j \rangle - \langle g_i, g_j \rangle|}{\|r_i\|\,\|r_j\|} + \left| \langle g_i, g_j \rangle \left( \frac{1}{\|r_i\|\,\|r_j\|} - \frac{1}{\|g_i\|\,\|g_j\|} \right) \right| \\
&\leq \frac{|\langle r_i, r_j \rangle - \langle g_i, g_j \rangle|}{\|r_i\|\|r_j\|} + \frac{|g_i^\top g_j|}{\|g_i\|\,\|g_j\|} \left| \frac{\|g_i\|\,\|g_j\| - \|r_i\|\,\|r_j\|}{\|r_i\|\,\|r_j\|} \right| \\
&\leq \underbrace{\frac{|\langle r_i, r_j \rangle - \langle g_i, g_j \rangle|}{\|r_i\|\,\|r_j\|}}_{A} + \underbrace{\left| \frac{\|g_i\|\,\|g_j\| - \|r_i\|\,\|r_j\|}{\|r_i\|\,\|r_j\|} \right|}_{B}
\end{aligned}
\tag{16}
$$

Using Cauchy-Schwarz inequality, the numerator of item $A$ is

$$
\begin{aligned}
|\langle r_i, r_j \rangle - \langle g_i, g_j \rangle| &= \left| (\mathbf{W}\boldsymbol{x}_i)^\top (\mathbf{W}\boldsymbol{x}_j) - (\widehat{\boldsymbol{y}}_i - \boldsymbol{y}_i)^\top (\widehat{\boldsymbol{y}}_j - \boldsymbol{y}_j) \cdot \boldsymbol{x}_i^\top \boldsymbol{x}_j \right| \\
&= \left| \boldsymbol{x}_i^\top \mathbf{W}^\top \mathbf{W} \boldsymbol{x}_j - \boldsymbol{x}_i^\top \boldsymbol{x}_j \cdot (\widehat{\boldsymbol{y}}_i - \boldsymbol{y}_i)^\top (\widehat{\boldsymbol{y}}_j - \boldsymbol{y}_j) \right| \\
&\leq \left| \boldsymbol{x}_i^\top \left[ \mathbf{W}^\top \mathbf{W} - (\widehat{\boldsymbol{y}}_i - \boldsymbol{y}_i)(\widehat{\boldsymbol{y}}_j - \boldsymbol{y}_j)^\top \right] \boldsymbol{x}_j \right| \\
&\leq (\sigma_1^2 + \epsilon^2)\|\boldsymbol{x}_i\|\|\boldsymbol{x}_j\|
\end{aligned}
\tag{17}
$$

The numerator of item $B$ is

$$
\begin{aligned}
|\|g_i\|\,\|g_j\| - \|r_i\|\,\|r_j\|| &\leq \|g_i\|\|g_j\| + \|r_i\|\|r_j\| \\
&\leq \|\widehat{\boldsymbol{y}}_i - \boldsymbol{y}_i\|\|\boldsymbol{x}_i\| \cdot \|\widehat{\boldsymbol{y}}_j - \boldsymbol{y}_j\|\|\boldsymbol{x}_j\| + \|\mathbf{W}\boldsymbol{x}_i\|\|\mathbf{W}\boldsymbol{x}_j\| \\
&\leq (\sigma_1^2 + \epsilon^2)\|\boldsymbol{x}_i\|\|\boldsymbol{x}_j\|
\end{aligned}
\tag{18}
$$

And,

$$\|r_i\|\|r_j\| \geq \sigma_2^2 \|\boldsymbol{x}_i\|\|\boldsymbol{x}_j\| \tag{19}$$

By combining these parts, we can have

$$|\Delta_{ij}| \leq 2\frac{\sigma_1^2 + \epsilon^2}{\sigma_2^2} \tag{20}$$

$\square$

Table 21: **Selection overlap** between gradient-based selection (LESS) and feature-based selection (ARDS) on ScienceQA. Higher overlap indicates stronger alignment between feature and gradient spaces.

| Selection Ratio | 30% | 40% | 50% | 60% | 70% |
|---|---|---|---|---|---|
| **Selection Overlap (%)** | 58.87 | 69.00 | 75.94 | 79.97 | 83.60 |

For empirical validation, we utilize the established gradient datastore in LESS and the conversation vector datastore in our ARDS to compute the overlap between gradient-based selection and our

feature-based selection for ScienceQA. As shown in Table 21, the two methods exhibit a high selection overlap, providing strong evidence that ARDS effectively approximates targeted instruction selection for robustness.

Guided by above theoretical insights and empirical evidence, we design a data-selection strategy for robust visual instruction tuning that scores samples by the cosine similarity of their conversation vectors, avoiding the heavy cost of gradient-based calculations.

