# OpenReview forum: "Data Selection Matters: Towards Robust Instruction Tuning of Large Multimodal Models"
_NeurIPS.cc/2025/Conference — NeurIPS 2025 poster_

### Official Review · Reviewer_NiSF · 2025-06-10

**Clarity:** 2
**Significance:** 3
**Originality:** 3
**Rating:** 4
**Confidence:** 3

**Summary:**

This paper propose a gradient-free robustness-aware data selection framework, a data filtering strategy that selects the worst-case samples based on embedding distances from a lightly fine-tuned proxy model, to improve robustness and efficiency for instruction tuning. The approach is evaluated across multiple multimodal and language understanding benchmarks, and the authors report that selecting approximately 30% of the data yields comparable or even better performance than using the full training set.

**Questions:**

See weakness above.
Overall, my main confusion with this paper lies in how the authors directly associate worst-case examples and accuracy improvements with robustness. I would appreciate further clarification on this point, and I’m open to engaging in further discussion.

**Ethical Concerns:**

["NO or VERY MINOR ethics concerns only"]

**Final Justification:**

The author's supplementary experiments solved my q2 and q3. Although I still have some doubts about q1, I think it can be improved to score 4.

**Limitations:**

yes

**Quality:**

3

**Strengths And Weaknesses:**

Strength:
1. The proposed ARDS method is conceptually simple yet well-motivated.
2. The method achieves consistently strong performance across a diverse set of benchmarks, highlighting the effectiveness of the proposed selection strategy.
3. The proposed data selection strategy is especially relevant for scenarios where computational resources are limited or training on full datasets is infeasible.

Weakness:
1. One major concern lies in the conceptual link between the data selection strategy and the robustness claims made throughout the paper. The authors heavily emphasize robustness as the central motivation, yet the core selection mechanism relies on embedding distances to identify worst cases. It is not sufficiently explained why samples that are semantically distant or harder to fit necessarily correspond to those that improve robustness. The assumption that worst-cases are aligned with robustness cases remains unsubstantiated, and the paper lacks an ablation or theoretical justification to support this connection. As a result, the method appears more as a form of relevance or difficulty-based sampling, rather than one explicitly targeting robustness.
2. I found the results on ARC-e and BoolQ in Table 3 puzzling, the full-data training performs significantly worse than using only a random 30% subset. This seems counterintuitive unless the full training set introduces severe biases that the smaller subset helps avoid. Could the authors elaborate on this surprising behavior? It would be helpful to include further analysis or visualization, such as label distributions, training loss curves, or predictions breakdown, to confirm this effect is genuine rather than due to setup discrepancies.
3. Figure 3 partially justifies the 30% ratio on two representative tasks, but it remains unclear whether ARDS consistently holds its advantage across tasks of varying modality complexity and reasoning depth. Given that the benchmark includes visual multiple-choice, text-based classification, and math QA, a brief discussion on whether this ratio remains effective across different cognitive requirements would strengthen the generality claim.
4. One point that remains underexplored is the practical motivation for data selection in the current setup. Since the full training data-LLAVA665k is assumed to be available, clean, and standard across benchmarks, it is not immediately clear why discarding a large portion (70%) of it would be desirable. The authors should clarify under what practical or deployment conditions such a selection strategy is intended to be applied. For example, is the method primarily motivated by training efficiency, overfitting concerns, or robustness to spurious correlations? Making this clearer would help situate the proposed method in real-world training scenarios.

---

> ### Author Rebuttal · Authors · 2025-07-31
>
> We would first like to express our genuine appreciation for the time and effort the reviewer has dedicated to reviewing our manuscript. We believe these comments and questions provide valuable feedback and will greatly help us improve the overall quality of our work. Please let us know whether our response has sufficiently addressed your concerns. We are more than happy to engage in further discussion.
>
> #### Q1.  Clarification of the association between worst-case selection and robustness gains
>
> > - We appreciate the opportunity to elucidate the connection of our ARDS and robustness goal and will incorporate the following discussion into our revised manuscript.
> >   - To begin, we would like to humbly highlight our key idea is to customize targeted instruction selection (e.g., LESS) for improving the robustness against specific dataset biases while avoiding gradient-based calculation.
> >   - **Targeted instruction selection improves robustness**.
> >       - **Theoretical Analysis.** Built on TracIn[1], LESS[2] quantifies the influence of a training datapoint $z$ on the loss of a test data $z^{\prime}$ via $\ell\left(z^{\prime} ; \theta^{t+1}\right)-\ell\left(z^{\prime} ; \theta^t\right) \approx -\eta_t\left\langle\nabla \ell\left({z} ; {\theta}^t\right), \nabla \ell\left(z^{\prime} ; \theta^t\right)\right\rangle$. Let $m$ denote vulnerable samples selected by our worst-case evaluation subgroups and $m^{\prime}$ denote perturbed test sample for robustness evaluation.
> >          - If the gradient inner-products $\left\langle\nabla \ell\left(m ; \theta^t\right), \nabla \ell\left(m^{\prime} ; {\theta}^t\right)\right\rangle$ is **notably non-negative**, then the loss of perturbed test samples $\ell\left(m^{\prime} ; \theta^{t+1}\right)-\ell\left(m^{\prime} ; \theta^t\right)$ decreases.
> >          - The **decreased loss mirrors adversarial training** for enhancing adversarial robustness [3].
> >       - **Empirical Validation.**
> >          - We first compute this gradient inner product using 100 training batches (batchsize=32) and 128 perturbed ScienceQA samples. The results of training batches from our robust training mixture show the average $\left\langle\nabla \ell\left(m ; {\theta}^t\right), \nabla \ell\left(m^{\prime} ; {\theta}^t\right)\right\rangle=25.76$, while that of training batches from random selection is $1.95$. This strongly supports our theoretical premise.
> >          - We kindly refer to Table 6 for the contribution of worst-case evaluation subgroups.
> >   - **Alignment between ARDS and targeted instruction selection**. To avoid the heavy cost of gradient-based calculations and sidestep the need for access to downstream data, we leverage the rich hidden representation of LMMs and construct worst-case evaluation subgroups on the training corpus.
> >     - **Theoretical Analysis.** In Appendix M, we provide a theoretical analysis under a linear model to demonstrate that the cosine similarity discrepancy between feature embeddings $r$ and gradients $g$ of two data samples $\left|\Delta_{i j}\right|=\left|\cos \left(r_i, r_j\right)-\cos \left(g_i, g_j\right)\right|$ can be upper-bounded.
> >     - **Empirical Validation.** We utilize the established gradient datastore in LESS and conversation vector datastore in our ARDS to compute the overlap between gradient-based selection and our feature-based selection for ScienceQA. The results, in the table below, demonstrate **high selection overlap**.
> >    - The **transitive relationship thus substantiates why ARDS enhances robustness**, supported by both theoretical insights and empirical evidence.
> >
> > | Selection ratio | 30% | 40% | 50% | 60% | 70%|
> > | --- | --- |--- |--- |--- |--- |
> > | Selection Overlap|58.87%|69.00%|75.94%|79.97%|83.60%|
> >
> > [1] Estimating training data influence by tracing gradient descent. In NIPS, 2020
> >
> > [2] Less: Selecting influential data for targeted instruction tuning. In ICML, 2024
> >
> > [3] Towards deep learning models resistant to adversarial attacks. In ICLR, 2018
>
> #### Q2. Why does full-data training perform worse than a random selection on pure-text tasks?
> > - We appreciate the reviewer's great feedback, and delve into a meticulous analysis for the decreased performance of full-data trained model on ARC-e and BoolQ.
> >     - We hypothesize that this phenomenon arises from the catastrophic forgetting driven by imbalanced training data across modalities, which aligns prior work that identified text-only forgetting during visual instruction tuning [1, 2] For example, WINGS [1] reports up to 13.33% text-only degradation after visual modality expansion.
> >     - In LLaVA-665K, only around 40688/665298 $\approx$ 6% of samples are pure-text instructions. During large‑scale multimodal tuning, the optimizer focuses overwhelmingly on vision‑centric tasks and gradually forgets language knowledge acquired from text‑only instruction tuning.
> >     - We have incorporated additional analysis by measuring text-only forgetting with random subsets of increasing size for LLaVA‑1.5 (7B), owing to compute limits. Larger subsets are expected to include more image-text pairs and thus amplify modality mismatch between multimodal and text-only inputs. The results, shown in table below, confirm that adding vision‑heavy data helps model excel in multimodal tasks while accentuates forgetting on text‑only tasks.
> >
> > |Task Type | Training Size | 5% | 10% | 30% | 50% | 70% | 90% | 100% |
> > | --- | --- | --- | --- | --- | --- | --- | --- | --- |
> > | text-only task |BoolQ Acc. | 64.10 | 58.44 | 55.93 | 53.43 | 55.66 | 48.35 | 37.77|
> > | multimodal task |A-OKVQA Acc. | 72.05 | 74.32 | 78.25 | 79.04 | 80.00 | 80.79 | 80.52 |
>
> > - We would like to humbly highlight that after once data selection, all subsequent results (e.g., ScienceQA, ARC and BoolQ) were obtained with the identical checkpoint trained with the selected data under the same evaluation setup.
> >
> > [1] Wings: Learning multimodal llms without text-only forgetting. In NIPS, 2024
> >
> > [2] Vila: On pre-training for visual language models. In CVPR. 2024
>
> #### Q3. More discussions concerning 30% ratio on other tasks
> > - We appreciate this great suggestion, and in response, we have incorporated additional experiments to assess the generalization of our curated robust training mixture with 30% ratio.
> >   - Specifically, we have, following your advice, included a comparison with other mentioned methods on another text-only benchmark (SocialIQA[1]) and two additional math benchmarks (MathVista[2] and DynaMath[3]). These two mathematical benchmarks can be viewed as more difficult Out-of-Domain benchmarks, as the original LLaVA-665K  does not explicitly involve math training data [4].
> >     - MathVista[2] is a mathematical reasoning benchmark introduced to analyze reasoning capabilities in visual complex scenarios (e.g., tables, function plots).
> >     - DynaMath[3] is a dynamic visual math benchmark designed to systematically analyze the robustness of mathematical reasoning capabilities across different topics (e.g., arithmetic, graph theory) under diverse condition changes (e.g., numerical value change). We report the average clean and robust accuracy on each benchmark.
> >   - The results, shown in the table below, demonstrate our curated robust training mixture with the 30% consistently boosts the generalization and robustness of visual instruction tuning, compared to full-data training and previous state-of-the-art selectors.
> >
> > | Method | data percentage | SocialIQA-Avg | MathVista-Avg | DynaMath-Avg |
> > | --- | --- | --- | --- | --- |
> > | Full |  | 48.70 | 23.38 | 24.48 |
> > | Random | 30% | 49.25 | 19.86 | 21.06 |
> > | LESS | 30% | 50.93 | 23.38 | 25.19 |
> > | COINCIDE | 30% | 52.30 | 18.98 | 19.98 |
> > | ARDS(ours) | 30% | 58.56 | 25.00 | 26.14 |
> >
> >
> > [1] Social IQa: Commonsense reasoning about social interactions. In EMNLP, 2019.
> >
> > [2] MathVista: Evaluating Mathematical Reasoning of Foundation Models in Visual Contexts. In ICLR, 2024
> >
> > [3] DynaMath: A Dynamic Visual Benchmark for Evaluating Mathematical Reasoning Robustness of Vision Language Models. In ICLR, 2025
> >
> > [4] Improved Baselines with Visual Instruction Tuning. In CVPR, 2024
>
> #### Q4. More in-depth discussions concerning the motivation
> > - We appreciate the reviewer's invaluable feedback.
> >   - We would like to humbly highlight our primary motivation is to improve the robustness of visual instruction tuning against specific dataset biases.
> >     - We agree with the reviewer that full training data-LLAVA665k are assumed to be clean, but **individual clean samples do not imply they will not reinforce spurious correlations or detrimental bias** [1]. Large multimodal models tend to memorize common concepts or prioritize stereotypes present in the training corpus [2]. For example, both "roses are red" and "banana is yellow" image-text pairs are clean samples, but training corpus dominated by these samples may make the model disregard the actual user input that deviates from those patterns, and thus harm generalization and robustness under distribution shift.
> >     - As shown both theoretically and empirically in Q1 discussion, our curated robust training mixture improves not only the robustness of visual instruction tuning but generalization capabilities as well—sometimes even exceeding the full-data baseline.
> >   - Beyond robustness, our smaller training mixture favorably enjoys data efficiency, which is also targeted by prior selection methods such as LESS and COINCIDE. The smaller training mixture is appealing for its **lightweight storage** requirements and **reduced training cost**. This makes our approach practical for real-world deployment.
> >
> > [1] An investigation of why overparameterization exacerbates spurious correlations. In ICML, 2020
> >
> > [2]  Holistic analysis of hallucination in gpt-4v (ision): Bias and interference challenges. arXiv preprint arXiv:2311.03287, 2023.

---

> > ### Comment · Reviewer_NiSF · 2025-08-02
> >
> > Thank you for the author's response, which has addressed some of my concerns. Including these additional experiments in the appendix would likely strengthen the paper further. I have updated my score to 4.

---

### Official Review · Reviewer_HSHf · 2025-06-26

**Clarity:** 3
**Significance:** 3
**Originality:** 3
**Rating:** 4
**Confidence:** 4

**Summary:**

This paper designs a novel visual instruction selection approach, named ARDS. ARDS first divides clusters based on the extracted embeddings of conversations and then calculate ranking scores for each visual instruction by taking both embedding similarity and loss into account. Since ARDS only need one forward process and no backward process for each instance, ARDS is faster than gradient-based competitive methods. Extensive comparisons and ablation experiments are conducted to validate the effectiveness of ARDS.

**Questions:**

1.	According to Equation 6, the value score of an instance is computed as an aggregation of the average cluster loss and the similarity between the instance and the cluster. Typically, there are some clusters with low average loss and others with significantly higher average loss. Clearly, instances that belong to or are closer to high-loss clusters tend to receive higher value scores. As a result, the selected samples are likely to be predominantly from these high-loss instances. My concern is that, in extreme cases, if the majority of selected samples come from such high-loss clusters, the model might partially or even largely forget the knowledge associated with low-loss clusters due to their underrepresentation in training. Therefore, I believe a statistical analysis of the selected samples across clusters is necessary, but the paper lacks this component.
2.	I do not understand how the ablation study in Table 6 is conducted. If no perturbation is applied, then x′ does not exist—how is Equation 4 computed in that case? And if clustering is not performed, how is the value score calculated? The paper lacks necessary clarification regarding this part of the experimental setup.

If my concerns are addressed, I would consider raising my rating.

**Ethical Concerns:**

["NO or VERY MINOR ethics concerns only"]

**Final Justification:**

The authors'rebuttal has addressed my concerns. After comprehensively assessing the paper's novelty and writing quality, I have decided to maintain the Rating of 4.

**Limitations:**

yes

**Quality:**

3

**Strengths And Weaknesses:**

Strengths:
1. The designed pipeline of ARDS is clear and intuitively meaningful. Authors make sufficient explorations in three key steps of ARDS: i) representation of conversation used for clustering; ii) identifying vulnerable samples as representations of clusters; iii) scoring mechinism that combines embedding similarity and average loss.
2. According to experimental results, ARDS outperforms competing methods across multiple benchmarks and requires less time consume.

Weaknesses:
1. The experimental section lacks an analysis of the distribution of the selected samples across the clusters.
2. The ablation settings are to some extent hard to understand.

For a more detailed explanation, please refer to the Questions section below.

---

> ### Author Rebuttal · Authors · 2025-07-31
>
> We sincerely thank the reviewer for providing valuable feedback. We detail our response below point by point. Please kindly let us know whether you have any further concerns.
>
>
> #### W1 & Q1: More analysis of selected samples across worst-case evaluation subgroups (clusters).
> > - We sincerely appreciate the reviewer's invaluable suggestion. Following your advice, we have incorporated the statistical analysis on the loss distribution of our worst-case evaluation subgroups.
> >   - Specifically, we measure balance through Shannon entropy given as $e(\ell _ {\mathcal{S}})=-\sum _ {m=0}^M p\left(\ell _ {\mathcal{S} _ m}\right) \log p\left(\ell _ {\mathcal{S} _ m}\right)$, where $p\left(\ell _ {\mathcal{S} _ m}\right)$ is the weight of the $m$-th cluster after $\text{Softmax}(\ell _ {\mathcal{S} _ m})$. As entropy depends on the number of subgroups, we further normalize entropy w.r.t. the number $M$ of possible clusters: $\bar{e}(\ell _ {\mathcal{S}})=\frac{e(\ell _ {\mathcal{S}})}{\log (M)}$, where $\log (M)$ is equal to the entropy of a uniform distribution of size $M$. We obtained the normalized entropy $\bar{e}(\ell _ {\mathcal{S}})=0.92$. The result empirically verifies that selected samples remain well dispersed under the loss-based weighted-sum aggregation strategy.
> > - We would like to humbly highlight that we only perform clustering for building the worst-case subgroups and the importance score of each training sample is aggregated from *all* subgroups in a weighted-sum manner. To validate that this choice mitigates imbalance, we replaced it with a max operator (i.e., the importance score of each datapoint is determined only by its *closest* cluster). The results, in Appendix Table 10, show that the weighted-sum aggregation strategy yields a better clean and robustness trade-off and avoids the skew that the max variant exhibits. We appreciate the reviewer for prompting this analysis and will highlight these results in our main paper.
>
>
>
> #### W2 & Q2: Clarification on the ablation settings in Table 6
> > - We apologize for any potential confusion and misunderstandings.
> >   - For the first row in Table 6, we randomly sample the same number of samples $MB$ as the final total size of the worst-case subgroups and the importance score of each training data is then aggregated with the sample-to-sample weighted-sum rule ( $\mathcal{I}\left(x_i\right)=\frac{\sum _ {j=1}^{MB} \exp \left(\ell_{j}\right) \cdot d _ {i j}}{\sum _ {j=1}^{MB} \exp \left(\ell _ j\right)}$ ). For the second row, we apply Equation 4 to retrieve top-$MB$ samples from the training dataset with the largest loss difference, and the importance score is aggregated in a sample-to-sample weighted-sum manner. The third row denotes our final construction strategy of worst-case subgroups, where we first perform clustering and use Equation 4 to retrieve top-$B$ candidates from each cluster. The importance score of training data is then aggregated with the sample-to-cluster weighted-sum scheme (Equation 6).

---

> > ### Comment · Reviewer_HSHf · 2025-08-03
> >
> > Thank you for your reply, which has effectively addressed my concerns. I was surprised to see that the normalized entropy value is 0.92, as this indicates that the distribution of the selected samples is relatively uniform. I suggest incorporating this statistical experiment into the paper, as it would greatly enhance the intuitive understanding of the sample distribution. Additionally, I acknowledge the contributions of this work. That said, given that the explanation of the ablation settings constitutes a crucial component of the experimental section—and that the original version lacks sufficient clarity in this regard—I will maintain the rating at 4.

---

### Official Review · Reviewer_LJnh · 2025-06-30

**Clarity:** 3
**Significance:** 2
**Originality:** 3
**Rating:** 4
**Confidence:** 4

**Summary:**

This paper proposes a data selection strategy for the SFT stage of MLLMs. The core idea is to utilize embeddings from a warmed-up proxy model to identify hard samples by measuring model consistency under random input variations, and then selecting data points around these identified hard samples.

**Questions:**

- How many augmented samples are generated for each seed instance during the selection process? What is the sensitivity of the proposed method's performance and computational cost to this number? A brief ablation study on the number of augmentations would be valuable.
- The analysis is primarily conducted on the SEED and ScienceQA benchmarks. Could the authors provide a more detailed justification for this specific choice?
- What is the performance of the proposed method on Vision-FLAN, the widely used dataset for selection?

**Ethical Concerns:**

["NO or VERY MINOR ethics concerns only"]

**Final Justification:**

Most of my concerns have been addressed or clarified with concrete results. My remaining concerns are about the fair presentation of attack-proof and attack-unknown work in the data selection field, as well as ensuring the code is made public, which the author has also addressed. Therefore, I will maintain my positive score.

**Limitations:**

The Limitation section currently reads more like a concluding summary and a brief outlook on future work. I recommend that the authors revisit the limitation section after expanding their experimental evaluation. This would allow for a much deeper and more valuable discussion.

**Quality:**

3

**Strengths And Weaknesses:**

+ It is encouraging to see that principles from active learning and data selection, such as coreset selection and consistency/uncertainty metrics [1, 2, 3], can be straightforwardly adapted and remain effective in the context of MLLMs.
- The manuscript would benefit from a more thorough discussion of its relationship to the broader field of active learning and data selection. Given the direct application of concepts from this area, explicitly positioning the work within this context would provide valuable perspective.
- The set of comparative methods is insufficient. The paper would be substantially strengthened by including comparisons against more recent and relevant baselines, such as D3M, which appears to be closely related, and ICONS [5], a direct extension of LESS. Furthermore, the evaluation should be expanded to include a wider range of standard MM benchmarks.
- A more detailed analysis of the time consumption is needed. This analysis should critically compare the proposed method's efficiency against strong baselines, particularly random sampling. And meantime take the performance gains into consideration (eg. comparison done by COINCIDE).
- Table 2, as the primary results table, is presented in a way that could be misleading. Following the established practice in MLLM data selection literature, the main results table should focus on performance comparisons against the full data pool and other selection methods under standard, unaltered evaluation benchmarks. It is unfair to evaluate competing methods on an augmented evaluation set where \textbf{the specific augmentations are leveraged by the proposed method but are unseen by the baselines}. The analysis of robustness should be presented in a separate table to ensure a fair and clear comparison of core performance.

[1]Coresets for Scalable Bayesian Logistic Regression
[2]Consistency-Based Semi-supervised Active Learning: Towards Minimizing Labeling Cost
[3]Not All Labels Are Equal: Rationalizing the Labeling Costs for Training Object Detection
[4]box-level active detection
[5]ICONS: Influence Consensus for Vision-Language Data Selection

---

> ### Author Rebuttal · Authors · 2025-07-31
>
> We would first like to express our genuine appreciation for the time and effort the reviewer has dedicated to providing such a detailed and in-depth review of our submission. We believe these comments and questions will contribute significantly to improving the overall quality of our work. We are more than happy to engage in further discussion.
>
> #### Q1. More discussions concerning the relationship to active learning and data selection.
> > - We sincerely thank the reviewer for the constructive comments and will incorporate the following discussion into our revised manuscript.
> >   - Both data selection and active learning aim to identify a subset of data that yields the best possible model, while the distinction lies in the focus on settings. Active learning defaults to that data is unlabled and emphasize on reducing annotation cost [1-5]. In contrast, data selection methods usually have full access to all labels when selecting [6], and the goal is to improve training efficiency [7] or robustness[8].
> >   - Our method ARDS is a data selection method with no extra annotation required. Our focus is to boost robustness against specific dataset biases for visual instruction tuning while avoiding computation-intensive gradient-based calculation in [7, 8].
>
> #### Q2. More comparisons with other related work and on a wider range of standard benchmarks.
> > - We appreciate the reviewer's great suggestion, and in response, we have conducted supplementary experiments to compare with D3M [8], which is a gradient-based data selection method designed for improving image classification robustness. Specifically, we reproduce D3M in our setting using the same warmed-up proxy model for extracting gradient vectors. We follow its standard procedure to project gradients, estimate the coefficient for each training sample and calculate the group alignment as the final score.
> >     - The results show that compared to this gradient-based method, the proposed ARDS achieves better robustness improvement while avoiding computation-intensive gradient calculation.
> >
> > | Method | data percentage | ScienceQA-Clean | ScienceQA-PA | ScienceQA-SA | ScienceQA-SA+PA | ScienceQA-Avg. |
> > | --- | --- | --- | --- | --- | --- | --- |
> > | Random | 30% | 69.76 | 52.60 | 59.44 | 23.75 | 51.39 |
> > | D3M | 30% | 69.86 | 56.87 | 62.52 | 27.17 | 54.10 |
> > | ARDS(ours) | 30% | 69.26 | 59.40 | 68.57 | 47.60 | 61.21 |
>
> > - Additionally, we have included more standard benchmarks for a more comprehensive comparison. The results, shown in the table below, demonstrate that our method achieves comparable generalization capability and consistently boosts robustness across various benchmarks. This further demonstrates the effectiveness of the proposed ARDS.
> >
> > | Method | data percentage | TextVQA | GQA | SocialIQA-Avg | MathVista-Avg | DynaMath-Avg |
> > | --- | --- | --- | --- | --- | --- | --- |
> > | Full |  | 57.23 | 61.94 | 48.70 | 23.38 | 24.48 |
> > | Random | 30% | 55.90 | 59.61 | 49.25 | 19.86 | 21.06 |
> > | LESS | 30% | 49.45 | 56.49 | 50.93 | 23.38 | 25.19 |
> > | COINCIDE | 30% | 56.09 | 59.15 | 52.30 | 18.98 | 19.98 |
> > | D3M | 30% | 54.41 | 56.73 | 51.28 | 23.69 | 20.55 |
> > | ARDS(ours) | 30% | 56.97 | 60.32 | 58.56 | 25.00 | 26.14 |
>
> #### Q3. More computational analysis with baselines
> > - We appreciate the reviewer's valuable suggestion, and in response, we have incorporated additional computational analysis.
> >    - We would first like to humbly emphasize that we use the standard LLaVA-1.5 training configurations when comparing all baseline methods, and thus the wall-clock time of visual instruction tuning on the same size of training mixture should be the same for all baseline methods.
> >    - While ARDS, together with prior competitor COINCIDE, extract sample features for information measure to perform data selection, which can indeed increase the computational cost compared to random selection, this trade-off is justified by our results significantly enhancing the robustness—a key motivation of our work.
> >
> > | Method | Selection Ratio | Wall-clock time of data selection | Wall-clock time of visual instruction tuning | SciPA-Avg. | SEED-Avg. |
> > | --- | --- | --- | --- | --- | --- |
> > | Random | 10% | <1min | 3.6h | 46.27 | 33.31 |
> > | COINCIDE | 10% | 100 min | 3.6h | 47.30 | 33.91 |
> > | ARDS (ours) | 10% | 39 min | 3.6h | 61.42 | 45.08 |
> > | Random | 30% | <1min | 6.9h | 51.39 | 37.97 |
> > | COINCIDE | 30% | 100 min | 6.9h | 52.27 | 39.58 |
> > | ARDS (ours) | 30% | 39 min | 6.9h | 61.21 | 46.80 |
>
> #### Q4. Split Table 2 into separate tables and add more results to ensure fair comparisons
> > - We appreciate the reviewer’s invaluable suggestion.
> >   - To begin, we would like to humbly emphasize that our primary motivation is to improve the robustness of visual instruction tuning against specific dataset biases. We also humbly emphasize that our method does not require access to downstream few-shot examples used in LESS or a reference model well-trained on the full data required by COINCIDE.
> >   - We will follow your advice to divide clean and robust accuracies in Table 2 into separate tables to make comparisons clearer.
> >   - Following your advice to ensure fair comparison, we have conducted additional experiments by adding our constructed worst-case evaluation subgroup as the selection source for the previous state-of-the-art selector COINCIDE. The results, shown in the table below, demonstrate the effectiveness of our ARDS to achieve superior robustness across benchmarks.
> >
> > | Method | Data Percentage | ScienceQA-Avg | SEED-Avg | MMBench-EN-Avg | A-OKVQA-Avg |
> > | --- | --- | --- | --- | --- | --- |
> > | COINCIDE | 30% | 52.27 | 39.58 | 59.54 | 64.97 |
> > | COINCIDE+worst-case subgroup | 30% | 53.21 | 40.97 | 60.445 | 67.77 |
> > | ARDS (ours) | 30% | 61.21 | 46.80 | 65.26 | 72.95 |
>
> #### Q5 Clarification and more results on the worst-case perturbation.
> > We deeply appreciate the reviewer’s feedback. To begin, we would like to humbly highlight that perturbations are introduced solely to build worst-case evaluation sub-groups, which are not applied during training. To build the worst-case subgroups, we inject task-specific visual and textual perturbations (i.e., symbol plus permutation attacks) simultaneously, and thus only one augmented sample is generated. We further explored the effect of generating multiple samples. The results, shown in the table below, demonstrate that the performance of the proposed method is not very sensitive to the number of augmentations.
> >
> > | #Aug. in Worst-case | Computational Cost | data selection ratio | ScienceQA-Clean | ScienceQA-PA | ScienceQA-SA | ScienceQA-SA+PA | ScienceQA-Avg |
> > | --- | --- | --- | --- | --- | --- | --- | --- |
> > | 1 | 39min  | 20% | 69.26 | 59.40 | 68.57 | 47.60 | 61.21 |
> > | 2 | 50min | 20% | 68.67 | 58.85 | 68.07 | 48.34 | 60.98 |
>
>
> #### Q6. Justification for our analysis of more evaluation benchmarks.
> > - We appreciate the reviewer's invaluable suggestion and we have incorporated additional, more detailed ablational analysis using A-OKVQA, MMMU, MMBench-EN, and MMBench-CN benchmarks. We report the average clean and robust accuracies on each benchmark. The results, in the table below, demonstrate that each component of the proposed method is consistently effective across benchmarks.
> >
> > | Worst-case Perturbation | Evaluation Subgroup Clustering | Data Percentage | ScienceQA-Avg. | SEED-Bench-Avg. | A-OKVQA-Avg. | MMMU-Avg. | MMBench-EN-Avg. | MMBench-CN-Avg. |
> > | --- | --- | --- | --- | --- | --- | --- | --- | --- |
> > | $\times$ | $\times$ | 30% | 52.39 | 40.43 | 66.03 | 20.92 | 60.53 | 55.42 |
> > | $\checkmark$ | $\times$ | 30% | 55.65 | 45.40 | 71.57 | 21.99 | 64.09 | 58.79 |
> > | $\checkmark$ | $\checkmark$ | 30% | 61.21 | 46.80 | 72.95 | 22.88 | 65.26 | 59.80 |
>
>
> #### Q7 More results on Vision-FLAN
> > - We sincerely appreciate the reviewer's invaluable suggestion and have conducted additional experiments on another training dataset Vision-Flan using the proposed ARDS. The results, shown in the table below, demonstrate that the robust training mixture created by ARDS on Vision-Flan achieves superior robustness, highlighting the effectiveness of our robustness-aware data selection method in mitigating the model's biased behaviors.
> >
> > | Method | Data Percentage | Sci-Clean | Sci-PA | Sci-SA | Sci-SA+PA | Sci-Avg. |
> > | --- | --- | --- | --- | --- | --- | --- |
> > | Full | 100% | 64.06 | 39.71 | 53.59 | 13.73 | 42.77 |
> > | Random | 50% | 61.38 | 34.41 | 53.59 | 18.89 | 42.06 |
> > | ARDS (Ours) | 50% | 62.87 | 37.98 | 60.54 | 33.56 | 48.74 |
>
> #### Q8. More discussion concerning limitations.
> > - We appreciate the reviewer's valuable feedback and will incorporate the following discussion of limitations in the revised manuscript. Although the efficacy of the proposed method has been confirmed through empirical experiments for mitigating several dataset biases, such as position bias, symbol-content spurious correlations, and visual spurious correlations, opportunities for refinement persist. One avenue for improvement involves a more diverse investigation into the potential dataset biases and their interaction effects.
>
> >
> > [1]Coresets for Scalable Bayesian Logistic Regression. In NIPS, 2016
> >
> > [2]Consistency-Based Semi-supervised Active Learning: Towards Minimizing Labeling Cost. In ECCV, 2020
> >
> > [3]Not All Labels Are Equal: Rationalizing the Labeling Costs for Training Object Detection. In CVPR, 2022
> >
> > [4]Box-level active detection. In CVPR, 2023
> >
> > [5]A Survey on Deep Active Learning: Recent Advances and New Frontiers, In TNNLS, 2024
> >
> > [6]A Survey on Data Selection for Language Models. In TMLR, 2024
> >
> > [7]Less: Selecting influential data for targeted instruction tuning. In ICML, 2024.
> >
> > [8]Improving subgroup robustness via data selection. In NIPS, 2024.

---

> > ### Comment · Reviewer_LJnh · 2025-08-07
> >
> > Thank  authors for their response and the effort put into the additional experiments. I have two concerns I would like to follow up on:
> >
> > Regarding W5, the stated goal is to select samples that are most robust to attack. A truly practical and meaningful validation of this would be to demonstrate robustness against novel, unknown attacks.
> > The current experimental setup, which uses an attack-aware method to compare against general selection baselines, creates an unfair comparison where the proposed method is essentially acting as both the player and the referee. In my opinion, a proper way to validate this setting would be to hold out a set of attacks or devise a new attack group for testing, rather than evaluating on a test where the challenges are known to the method beforehand.
> >
> > Additionally, I would like to inquire if the authors intend to release the code for reproducibility.

---

> > > ### Author Response · Authors · 2025-08-09
> > >
> > > We appreciate the reviewer for reviewing our rebuttal and for allowing us to further clarify your concerns.
> > >
> > >
> > >
> > > #### Concern 1: More in-depth discussions concerning attack awareness during data selection
> > > > - We appreciate the reviewer's insightful suggestion to further evaluate our proposed data selection setting on **novel attacks**.
> > > >   - We first kindly refer to **Appendix Table 13** for the robustness evaluation on ScienceQA under **previously unseen attacks**, where we introduced two previously unseen symbol attacks and an additional visual corruption.
> > > >   - We have conducted additional comparisons on more benchmarks to **extend Appendix Table 13**, pitting ARDS against Random selection, COINCIDE, and its strengthened variant—COINCIDE supplied with our worst-case subgroup to fully align the seen data during selection.
> > > >      - **SA1** and **SA2** denote the canonical answer labels A/B/C/D are replaced by S/N/V/F and U/I/O/P, respectively, which are injected as **novel variations** not encountered during data selection (recall we used Q/W/E/R to build worst-case evaluation subgroups)
> > > >      - **PA** means the adversarial permutation attack, **adaptively generated** for each new question by enumerating all possible permutations of the answer options.
> > > >      - The results, presented in table below, demonstrate that **ARDS consistently achieves strong robustness and generalization even in the presence of these new, previously unseen attacks**. Even when compared with the COINCIDE+worst-case subgroup, ARDS maintains superior robust accuracy across all scenarios, **confirming that the robustness gains generalize well beyond attacks seen during data selection.**
> > > > - We would also like to humbly highlight that **the setting of defender knowing the attack type of adversary aligns with standard practices in adversarial training and evaluation**, where models are usually trained against specific adversarial perturbation budget, such as $l_{\infty}$ [1], $l_{2}$[2], $l_{1}$[3]-norm bounded perturbations, and later evaluated on the corresponding type of attacks to measure robustness improvement over clean training.
> > > > - We appreciate the reviewer's invaluable feedback again, which has strengthened our work.
> > > >
> > > > | Selection Method | Data Percentage | Sci.-SA1 | Sci.-SA1+PA | Sci.-SA2 | Sci.-SA2+PA | **Sci.-Avg.** | SEED-SA1 | SEED-SA1+PA | SEED-SA2 | SEED-SA2+PA | SEED-Avg. | A-OKVQA-SA1 | A-OKVQA-SA1+PA | A-OKVQA-SA2 | A-OKVQA-SA2+PA | A-OKVQA-Avg. | MMBench-SA1 | MMBench-SA1+PA | MMBench-SA2 | MMBench-SA2+PA | MMBench-Avg. |
> > > > | --- | --- | --- | --- | --- | --- | --- | --- | --- | --- | --- | --- | --- | --- | --- | --- | --- | --- | --- | --- | --- | --- |
> > > > | Full | 100% | 67.72 | 40.80 | 66.73 | 36.49 | 52.93 | 56.62 | 25.47 | 55.04 | 19.88 | 39.25 | 77.21 | 52.40 | 76.07 | 48.03 | 63.42 | 69.05 | 46.29 | 68.91 | 36.61 | 55.21 |
> > > > | Random | 30% | 64.06 | 32.92 | 63.81 | 31.28 | 48.01 | 47.82 | 13.57 | 47.22 | 8.29 | 29.22 | 60.26 | 24.98 | 65.15 | 18.34 | 42.18 | 60.18 | 29.36 | 62.51 | 21.30 | 43.33 |
> > > > | COINCIDE | 30% | 65.44 | 37.04 | 63.41 | 29.85 | 48.94 | 51.65 | 19.40 | 48.58 | 11.82 | 32.86 | 68.47 | 37.03 | 66.55 | 25.68 | 49.43 | 64.22 | 35.27 | 64.29 | 23.98 | 46.94 |
> > > > | COINCIDE+worst-case subgroup | 30% | 62.87 | 33.96 | 63.16 | 31.04 | 47.75 | 50.28 | 16.19 | 51.08 | 16.03 | 33.39 | 67.34 | 29.26 | 71.79 | 35.90 | 51.07 | 63.76 | 30.49 | 65.90 | 26.31 | 46.62 |
> > > > | ARDS (Ours) | 30% | **68.57** | **48.34** | **66.14** | **39.76** | **55.70** | **57.48** | **33.81** | **54.95** | **22.27** | **42.13** | **78.60** | **63.67** | **74.50** | **49.00** | **66.44** | **71.75** | **51.98** | **70.48** | **42.50** | **59.18** |
> > > >
> > > > [1] Towards deep learning models resistant to adversarial attacks. In ICLR, 2018
> > > >
> > > > [2] Adversarial training and robustness for multiple perturbations. In NIPS, 2019
> > > >
> > > > [3] Towards stable and efficient adversarial training against l1 bounded adversarial attacks. In ICML, 2023
> > >
> > >
> > > #### Concern 2: Additionally, I would like to inquire if the authors intend to release the code for reproducibility.
> > > > Yes. To promote transparency and reproducibility, we will release the source code.

---

> > > > ### Comment · Reviewer_LJnh · 2025-08-09
> > > >
> > > > Thank the authors for additional experiments and analysis, which partly addressed my concerns. I understand that dealing with known attack is a standard practice in adversarial training. However, for MLLM data selection methods, this direction of research is relatively new and unknown to the design process of previous competitors. While I recognize its value, Id like to make sure the comparison is presented as fair as possible. I will maintain my positive score.

---

### Official Review · Reviewer_5LuU · 2025-07-01

**Clarity:** 3
**Significance:** 2
**Originality:** 2
**Rating:** 4
**Confidence:** 4

**Summary:**

This paper focuses on developing data selection method to alleviate the data bias and supious correlation in the visual instruction tuning data mixture. The paper is well-written and the experimental section is well-organized. The proposed method ARDS is evaluated in serveral aspects and many benchmarks.

**Questions:**

See weakness.

**Ethical Concerns:**

["NO or VERY MINOR ethics concerns only"]

**Final Justification:**

After reviewing the authors’ responses to the weaknesses I raised, I am satisfied that they have fully addressed my concerns. I continue to regard this work as highly valuable and will maintain my positive score. Moreover, I hope this discussion will inspire future research to develop methods that can be applied without access to the entire dataset for data selection, an approach better suited to real‐world scenarios, where data and knowledge are constantly evolving.

**Limitations:**

Please add the limitation discussion at the end of paper.

**Paper Formatting Concerns:**

Add limitation section.

**Quality:**

3

**Strengths And Weaknesses:**

Pros:
- The proposed method is gradient-free.
- The paper is well-organized.
- The paper evaluates the ARDS extensively across various benchmarks and settings.

Cons:
- The main concern is that in a real scenario, one cannot obtain the whole training dataset at one time, if new data comes, one should recalculate the worst-case evaluation groups again, which would cost a lot. Ideally, we would like a data selection method that could directly give a decision on each candidate data before training and inference.

- If you compress the 30% data again, how small will the final data mixture be?

Small typo errors: There is no Figure 4 in the main paper. Figure 4 in Line 271 cannot be clicked.

---

> ### Author Rebuttal · Authors · 2025-07-31
>
> Thank you sincerely for your thoughtful feedback on our work. Below, we have provided a detailed explanation for your concerns as follows. Please do not hesitate to let us know if you have any further questions.
>
> #### Q1. More results in scenarios where the whole training dataset is not available.
> > - We appreciate the reviewer bringing up the concern in a scenario where one cannot obtain the whole training dataset at one time.
> >     - We would first like to highlight that our experimental setting follows existing static visual instruction selection methods, where the selectors are aware of all training data to build a gradient database [1] or perform clustering on the whole dataset [2]. The key idea of our worst-case evaluation subgroups is to find representative samples that are most vulnerable to model-biased behavior while not needing the downstream few-shot examples required in [1].
> >     - We totally agree with the reviewer that a selection method supporting dynamically coming data is desired. To further address the concern about the training data requirement for building our worst-case evaluation subgroups, we have conducted supplementary experiments. We **randomly sample 10% training data as the obtained small subset** of the original training corpus and **treat the remaining 90% as new incoming training data**. We denote the variant as ARDS$^*$. Specifically, to build worst-case evaluation subgroups, we perform clustering on these 10% data with the same number of clusters $K$ and subgroup budget $B$ and then apply task-specific perturbations (see Appendix Line 424-428 for more details). The results indicate that despite seeing only a tenth of the corpus during subgroup construction, ARDS$^\*$ achieves robustness close to the full-information ARDS—and still outperforms both LESS and COINCIDE. This highlights the **potential and applicability of our method to more dynamic data selection scenarios.**
> >
> > |Method|Selection Ratio | Sci-Clean | Sci-PA | Sci-SA | Sci-PA+SA | Sci-Avg. |
> > | --- | --- | --- | --- | --- | --- | --- |
> > |LESS [1]     | 30% | 68.42 | 55.63 | 64.70 | 34.95 | 55.93 |
> > |COINCIDE [2] | 30% | 67.72  | 52.21 | 61.08 | 28.06 | 52.27 |
> > |ARDS | 30% | 69.26 | 59.40 | 68.57 | 47.60 | 61.21 |
> > |ARDS$^*$ | 30% | 68.86 | 58.30 | 67.13 | 42.39 | 59.17 |
> >
> > [1] Less: Selecting influential data for targeted instruction tuning. In ICML, 2024.
> >
> > [2] Concept-skill transferability-based data selection for large vision-language models. In EMNLP, 2024.
>
>
> #### Q2. More results on further data selection.
> > - We appreciate the reviewer's insightful suggestion regarding performing data selection a second time.
> >   - To begin, we would like to humbly emphasize our current 30% ratio is determined by the trade-off between data efficiency as well as both clean and robust performance across varying tasks (kindly refer to Appendix Figure 5)
> >   - Following your advice, we have incorporated additional experiments to explore the impact of performing data selection again. The total selection ratio equals to the product of the two selection ratios. The results, in the table below, demonstrate that iterative selection, which first chooses 30% and then selects that subset to 33.3%, outperforms a single 10% selection. This indicates the great potential of our ARDS method to be extended to other scenarios. We will add the results and the discussion to our revised manuscript.
> >
> > | Method | Total Selection Ratio | 1st Selection Ratio | 2nd Selection Ratio | Sci-Clean | Sci-PA | Sci-SA | Sci-PA+SA | Avg. |
> > | --- | --- | --- | --- | --- | --- | --- | --- | --- |
> > | Random      | 10% | 10% | - | 66.48 | 45.31 | 55.68 | 17.60 | 46.26 |
> > | COINCIDE    | 10% | 10% | - | 66.24 | 48.14 | 56.57 | 18.24 | 47.30 |
> > | ARDS (ours) | 10% | 10% | - | 68.72 | 55.73 | 68.86 | 52.35 | 61.42 |
> > | ARDS (ours) | 10% | 30% | 33.33% | 71.10 | 60.14 | 71.15 | 56.42 | 64.70 |
>
>
> #### Q3. Typos correction
> > We appreciate the reviewer for pointing out these typos. We have revisited and corrected the typos in our manuscript.
>
>
> #### Q4. Clarification of limitation discussion
> > We appreciate the reviewer's feedback, and we kindly refer to Appendix A for the limitation discussion.

---

> > ### Comment · Reviewer_5LuU · 2025-08-06
> >
> > Thank you to the authors for their response.
> >
> > Regarding Weakness 1, I totally agree with the current setting. However, I would also suggest exploring a novel method to address the data‐selection problem without full knowledge of the entire training set, as I believe this more closely reflects real-world scenarios.
> >
> > For Weakness 2, I appreciate the additional experiments you’ve conducted. They address my curiosity about iterative data selection, and the results are intriguing: two-stage selection outperforms one-stage selection at the same overall ratio. It would be valuable to see an experiment in which you select 50% (just an example) of the data in the first stage, then an additional XX % in the second stage, resulting in a total of 30 % selected, and compare this to the baseline 30 % one-stage selection. Such an analysis would further strengthen the impact of your paper.
> >
> > Overall, I find this work very valuable and will maintain my positive score.

---

> > > ### Author Response · Authors · 2025-08-09
> > >
> > > #### More results on extending our method to iterative data selection.
> > > > We appreciate the reviewer’s invaluable suggestion and have conducted additional experiments that compare one-stage and two-stage iterative selection with ARDS while keeping the total selection ratio at 30%.
> > > > - Specifically, we vary the stage-1 ratio in {40%,50%,60%} and set the stage-2 ratio to keep the same total selection ratio.
> > > > - We observe that performance depends on how the 30% budget is split: the 60%→50% configuration yields a **comparable** performance to one-stage 30%, while other ratio allocations are slightly lower. Compared to the benefits shown in our earlier iterative selection results (30%→33.3% outperforms a single 10% selection), this also suggests that iterative selection gains might be saturated when the total selection ratio is relatively large. In addition, we believe exploring improved ratio budget splits for iterative selection deserves future work.
> > > >
> > > > | Method | Total Selection Ratio | 1st Selection Ratio | 2nd Selection Ratio | Sci-Clean | Sci-PA | Sci-SA | Sci-PA+SA | Avg. |
> > > > | --- | --- | --- | --- | --- | --- | --- | --- | --- |
> > > > | Random | 30% | 30% | - | 69.76 | 52.60 | 59.44 | 23.75 | 51.39 |
> > > > | COINCIDE | 30% | 30% | - | 67.72 | 52.21 | 61.08 | 28.06 | 52.27 |
> > > > | ARDS (ours) | 30% | 30% | - | 69.26 | 59.40 | 68.57 | 47.60 | 61.21 |
> > > > | ARDS (ours) | 30% | 40% | 75% | 69.51 | 57.46 | 66.44 | 40.11 | 58.38 |
> > > > | ARDS (ours) | 30% | 50% | 60% | 69.16 | 57.26 | 66.39 | 41.75 | 58.64 |
> > > > | ARDS (ours) | 30% | 60% | 50% | 69.46 | 58.06 | 67.77 | 47.15 | 60.61 |

---

### Author Response · Authors · 2025-08-02
**Summary**

We extend our sincere thanks to the reviewers for their constructive feedback. In order to provide greater clarity on the revisions made to our paper and the experiments we conducted to address the reviewers' questions, we have summarized the experiments and clarifications made during the rebuttal period as follows.

**Additional Experiments**
- Extended ARDS to scenarios where the complete training set is unavailable (Reviewer 5LuU Q1)
- Extended ARDS to a second-round selection. (Reviewer 5LuU Q2)
- Added comparisons to more related work and on a wider range of benchmarks. (Reviewer LJnh Q2, Reviewer NiSF Q3)
- Provided a more detailed comparative analysis of the computational overhead. (Reviewer LJnh Q3)
- Applied our proposed worst-case evaluation subgroups to prior selection methods for a direct comparison. (Reviewer LJnh Q4)
- Analyzed how performance varies with the number of generated samples when building worst-case subgroups. (Reviewer LJnh Q5)
- Performed additional ablations on more benchmarks. (Reviewer LJnh Q6)
- Extended ARDS to other training datasets. (Reviewer LJnh Q7)
- Analyzed the loss distribution across worst-case evaluation subgroups. (Reviewer HSHf Q1)
- Analyzed the observed catastrophic forgetting on pure-text tasks. (Reviewer NiSF Q2)


**Clarifications**
- Added a high-level theoretical analysis of ARDS. (Reviewer NiSF Q1)
- Added a deeper discussion of our core motivation (Reviewer NiSF Q4)
- Clarified experimental settings used in ablations. (Reviewer HSHf Q2)
- Added a deeper discussion of the ARDS's position within active learning and data selection areas. (Reviewer LJnh Q1)
- Added a deeper limitation discussion (Reviewer LJnh Q8)
- Corrected typos (Reviewer 5LuU Q3)

---

### Note · Authors · 2025-08-15

Dear Area Chairs and Reviewers,

We sincerely appreciate your time and efforts in handling our submission, which have greatly strengthened our work. To support your final assessment, we have prepared an overview of our discussions.
- We are glad that we addressed all reviewer questions and concerns during rebuttal and discussion, leading to a consensus recommendation for acceptance.
- In the initial reviewes, we are delighted to note that reviewers found our data selection method "**gradient-free**” (`Reviewer 5LuU`), "**intuitively meaningful**" (`Reviewer HSHf`), "**well-motivated**" (`Reviewer NiSF`), and "**supported by extensive experiments**” (`Reviewer 5LuU`, `Reviewer HSHf` `Reviewer NiSF`), with “**clear**" (`Reviewer HSHf`) / "**well-organized**” (`Reviewer 5LuU`) writing.
- Reviewers suggested strengthening the work with additional experiments and analyses, including extensions to practical scenarios (Reviewer `5LuU`), broader comparisons across more benchmarks (Reviewer `LJnh`, Reviewer `NiSF`) and evaluations under novel attacks (Reviewer `LJnh`). We have incorporated all of these requests, demonstrating that our method ARDS can be extended to scenarios without access to the full training dataset or second-round selection, and **ARDS consistently outperforms baselines under novel attacks and across more challenging benchmarks, including visual mathematical reasoning robustness benchmarks**.
- Beyond these experimental additions, we also conducted statistical analysis of the distribution of proposed worst-case subgroups (Reviewer `HSHf`), and provided **high-level theoretical insights supported by empirical validation, clarifying the connection between worst-case selection and robustness improvements** (Reviewer `NiSF`).


We are deeply grateful for the reviewers' valuable suggestions, and we will include these improvements in our final manuscript.

---

### Decision · Program_Chairs · 2025-09-17

**Decision:**

Accept (poster)

**Comment:**

This paper proposes a visual instruction selection approach, named ARDS, for the SFT stage of MLLMs. The core idea is to utilize embeddings from a warmed-up proxy model to identify hard samples by measuring model consistency under random input variations, and then selecting data points around these identified hard samples. As ARDS only needs one forward process and no backward process for each instance, it is faster than gradient-based methods. However, it is also somewhat costly due to its heavy pipeline, which involves hierarchical clustering, perturbation, and loss difference evaluations. Extensive comparisons and ablation experiments are conducted to validate the effectiveness of ARDS.

The paper received unanimously positive feedback from the reviewers. Most of the reviewers' concerns have been addressed during the rebuttal phase. I encourage the authors to include the supplementary experiments as well as some of the discussions during the rebuttal in their final version.